# Accelerating Feature Conformal Prediction
# via Taylor Approximation

**Zihao Tang**[⋆]  **Boyuan Wang**[⋄]  **Chuan Wen**[∘]  **Jiaye Teng**[⋆†]

[⋆]Shanghai University of Finance and Economics
[⋄]Southern University of Science and Technology
[∘]Shanghai Jiao Tong University

## Abstract

Conformal prediction is widely adopted in uncertainty quantification, due to its post-hoc, distribution-free, and model-agnostic properties. In the realm of modern deep learning, researchers have proposed Feature Conformal Prediction (FCP), which deploys conformal prediction in a feature space, yielding reduced band lengths. However, the practical utility of FCP is limited due to the time-consuming non-linear operations required to transform confidence bands from feature space to output space. In this paper, we present Fast Feature Conformal Prediction (FFCP), a method that accelerates FCP by leveraging a first-order Taylor expansion to approximate these non-linear operations. The proposed FFCP introduces a novel non-conformity score that is both effective and efficient for real-world applications. Empirical validations showcase that FFCP performs comparably with FCP (both outperforming the Split CP version) while achieving a significant reduction in computational time by approximately 50x in both regression and classification tasks. The code is available at `https://github.com/ElvisWang1111/FastFeatureCP`.

## 1  Introduction

Machine learning has been successfully applied in numerous fields such as computer vision, natural language processing, and gaming [Jordan and Mitchell, 2015, Silver et al., 2017]. However, machine learning models usually suffer from overconfidence issues [Wei et al., 2022] and even hallucinations in large language models (LLMs) [Ji et al., 2023], which makes them unreliable and unable to be deployed in fields like finance and medicines [Gelijns et al., 2001, Thirumurthy et al., 2019, Morduch and Schneider, 2017]. Therefore, it is essential to develop techniques for uncertainty quantification and calibrate the original machine learning models Abdar et al. [2021], Guo et al. [2017], Chen et al. [2021], Gawlikowski et al. [2021].

Among the uncertainty quantification techniques, Conformal Prediction (Split CP, or split conformal prediction, Vovk et al. [2005]; Shafer and Vovk [2008b]; Burnaev and Vovk [2014]) stands out, because it is distribution-free, does not require retraining, and can be directly applied to various models. Conformal prediction deploys a calibration step to calibrate a base model and then construct the confidence band. The goal of conformal prediction is to return a band $\mathcal{C}_{1-\alpha}(X')$ such that

$$\mathbb{P}(Y' \in \mathcal{C}_{1-\alpha}(X')) \geq 1 - \alpha, \tag{1}$$

where $(X', Y')$ denotes a test point and $1 - \alpha$ represents the confidence level.

In deep learning regimes, researchers try to utilize feature information in Split CP, since the feature space usually contains meaningful semantic information in neural networks [Shen et al., 2014]. This leads to Feature Conformal Prediction (FCP, Teng et al. [2022]).

---

[†]Correspondence to `tengjiaye@sufe.edu.cn`

39th Conference on Neural Information Processing Systems (NeurIPS 2025).

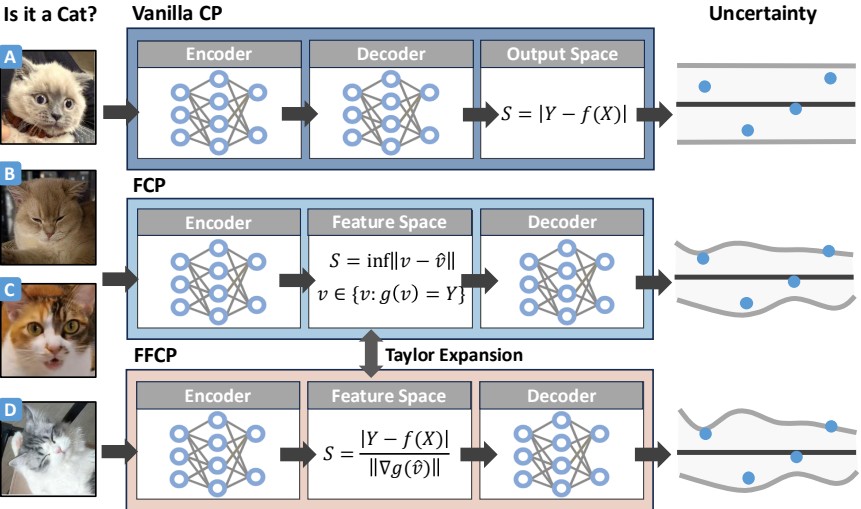

Figure 1: Comparison among Split CP, FCP, and FFCP. FCP and FFCP are more efficient compared to Split CP since they return different band lengths for different individuals. This is achieved by calculating a non-conformity score in the feature space. Besides, FFCP approximates FCP using a Taylor expansion, which leads to a different non-conformity score and accelerates the transformation from feature space to output space.

Fortunately, one may get different band lengths on different individuals by utilizing the feature information, leading to a shorter confidence band. As a comparison, Split CP only returns the same band length for all individuals in regression tasks, which indicates a longer length.

However, the practical applications of FCP are limited because (a) it is time-consuming, and (b) it only returns *estimated* bands on the output space, making it less efficient. These two issues come from the step *Band Estimation*, which transfers the confidence band from feature space to output space. This step involves complex non-linear operations called LiPRA [Xu et al., 2020] and therefore (a) the non-linear operation requires high computational costs, and (b) the configurations in LiPRA might finally influence the estimated band, further harming the performance of FCP.

In this paper, we present Fast Feature Conformal Prediction (FFCP), which offers a fast version for handling the aforementioned nonlinear operations in FCP. Different from Split CP and FCP, FFCP introduces a novel non-conformity score $s_{\text{ff}}(\cdot)$ that is simple to compute and does not require additional training,

$$s_{\text{ff}}(X, Y, g \circ h) = |Y - g \circ h(X)| / \|\nabla g(\hat{v})\|, \tag{2}$$

where $(X, Y)$ denotes a sample, $g \circ h$ denotes a neural network with a feature layer $h$ and a prediction head $g$, and the gradient $\nabla g(\hat{v})$ denotes the gradient of $g(\cdot)$ on the trained feature $\hat{v} \triangleq h(X)$, namely, $\nabla g(\hat{v}) = \frac{\mathrm{d}g \circ h(X)}{\mathrm{d}h(X)}$. We refer to Algorithm 2 for more details and illustrate the algorithm in Figure 1.

The above non-conformity score is closely related to FCP. Specifically, **FFCP with this non-conformity score can be regarded as a fast version of FCP, since it equivalently approximates the prediction head using a Taylor expansion**, which simplifies the aforementioned nonlinear operations. Fortunately, FFCP inherits the merits of FCP, for example, it also utilizes the semantic information in the feature.

From a theoretical perspective, we first demonstrate that FFCP is effective in Theorem 4.1, in that it returned a confidence band with empirical coverage larger than the given confidence $1 - \alpha$. Additionally, we demonstrate in Theorem 4.2 that FFCP produces a shorter confidence band than Split CP under a proposed square condition. The square conditions outline the properties of the feature space from two perspectives: expansion and quantile stability, implying that the feature space has a smaller distance between individual non-conformity scores and their quantiles. This reduces the cost of the quantile operation and therefore leads to a shorter confidence band. We also validate the square conditions using empirical observations.

From an empirical perspective, we conduct several experiments on real-world datasets and show that FFCP performs comparably with FCP, both outperforming Split CP, **while achieving nearly 50 times the speed of FCP in terms of runtime** for regression tasks. We further validate the approximation ability of FFCP with FCP using the correlation between the non-conformity score of FFCP and FCP. We also apply FFCP to the image segmentation problems to verify its general applications. Besides, we show that the concept in FFCP is pretty general, and can be combined with other variants of CP, *e.g.*, CQR [Romano et al., 2019a] and LCP [Guan, 2023] in regression tasks, and RAPS [Angelopoulos et al., 2020] in classification tasks.

Overall, our main contributions are summarized as follows:

- This work proposes FFCP, which serves as a fast version of FCP. FFCP achieves around 50x faster speed compared to FCP (Table 1) by utilizing Taylor expansions to approximate the prediction head in FCP. Besides, FFCP inherits the merits of FCP and efficiently exploits semantic information in the feature space.

- Theoretical insights demonstrate that FFCP returns shorter band length compared to Split CP (Theorem 4.2) while ensuring coverage exceeds the given confidence level under mild conditions (Theorem 4.1).

- Extensive experiments with both synthetic and real data demonstrate the effectiveness of the proposed FFCP algorithm (Table 2). Additionally, we demonstrate the universal applicability of our gradient-level techniques by extending them to other tasks such as classification (FFRAPS, Algorithm 5) and segmentation, and to various conformal prediction variants, including CQR (Algorithm 3) and LCP (Algorithm 4).

## 2  Related Work

Conformal prediction is a post-hoc calibration method dealing with uncertainty quantification [Vovk et al., 2005, Shafer and Vovk, 2008a, Barber et al., 2020], which is deployed in numerous fields [Ye et al., 2024, Kumar et al., 2023, Quach et al., 2023]. The variants of conformal prediction typically revolve around the concept of non-conformity scores, with four main branches of development.

**Relaxing Exchangeability.**  The first branch relaxes the exchangeability requirement [Tibshirani et al., 2019, Hu and Lei, 2020, Podkopaev and Ramdas, 2021, Barber et al., 2022], leveraging weighted or reweighted quantiles to relax exchangeability. By doing so, it gains more flexibility and broader applicability in handling data that may not satisfy the standard exchangeability assumptions.

**Diverse Structures.**  The second branch applies conformal prediction to various data structures, for example, classification tasks [Romano et al., 2020, Angelopoulos et al., 2020], time series data [Xu and Xie, 2021, Gibbs and Candès, 2021], censored data in survival analysis [Teng et al., 2021, Candès et al., 2023], high-dimensional data [Candès et al., 2021, Lei et al., 2013], Bellman-based data [Yang et al., 2024], counterfactuals and individual treatment effects [Lei and Candès, 2021], *etc.*

Another way involves model structures, such as $k$-NN regression [Papadopoulos et al., 2011a], quantiles incorporated [Romano et al., 2019b, Sesia and Candès, 2020], density estimators [Izbicki et al., 2020b], and conditional histograms [Sesia and Romano, 2021]. These methods further enrich the application scenarios of conformal prediction by adapting it to diverse model frameworks.

**Enhancing Methods.**  The third branch focuses on enhancing the original conformal prediction with band length. Izbicki et al. [2020a] introduce CD-split and HPD-split methods, and Yang and Kuchibhotla [2021] develop selection methods to minimize band length. Of particular note is feature conformal prediction [Teng et al., 2022], which leverages neural network training information via feature spaces to improve band length.

**Localized Conformal Prediction.**  The fourth branch focuses on enhancing the non-conformity score normalization by incorporating difficulty-related terms like $\frac{\|Y-\hat{Y}\|}{\sigma(X')}$ where $Y$ denotes the true label, $\hat{Y}$ denotes the predicted label, and $\sigma(X')$ denotes the standard deviation related to $X'$. Here are three key approaches:

*(1) Weight Adjustment via Calibration Distances.* This approach calculates the distance from the test point to the calibration points and then uses these distances to define the weights of non-conformity scores in the calibration process [Han et al., 2022, Guan, 2023]. Our gradient-level techniques can be used to combine with this branch (see FFLCP in Algorithm 4).

*(2) Normalization Using Proximity to Training Set.* This approach utilizes the observation that a testing point exhibits smaller uncertainty when it is close to the training set, and uses such metrics to approximate $\sigma(X')$ [Papadopoulos et al., 2008, 2011b, Papadopoulos and Haralambous, 2011]. In the deep learning regimes, we believe that such procedures can be further improved by calculating the distance in the feature space rather than the input space, since feature layers usually contain more semantic information.

*(3) Modeling $\sigma(X')$.* This approach trains a separate model to estimate $\sigma(X')$, thereby enhancing the accuracy and adaptability of CP [Seedat et al., 2023, 2024]. However, this line of work heavily relies on the model performance and incurs high computational costs. As a comparison, our approach does not require additional training procedures.

**Uncertainty Quantification.** Uncertainty quantification is one of the most fundamental questions in machine learning. In addition to conformal prediction, many other approaches exist for quantifying uncertainty, including calibration-based techniques [Guo et al., 2017, Kuleshov et al., 2018, Nixon et al., 2019, Abdar et al., 2021, Chang et al., 2024] and Bayesian-based techniques [Blundell et al., 2015, Hernández-Lobato and Adams, 2015, Li and Gal, 2017, Izmailov et al., 2021, Jospin et al., 2022].

## 3 Preliminaries

We begin by introducing a dataset $\mathcal{D} = \{(X_i, Y_i)\}_{i \in [n]}$ indexed by $\mathcal{I}$. We split the dataset into two folds: a training fold $\mathcal{D}_{\text{tra}}$ indexed by $\mathcal{I}_{\text{tra}}$, and a calibration fold $\mathcal{D}_{\text{cal}}$ indexed by $\mathcal{I}_{\text{cal}}$. Denote the testing point by $(X', Y')$. For the model part, define $f$ as a neural network. We partition $f = g \circ h$, where $h$ denotes the feature function (the initial layers of the neural network) and $g$ denotes the prediction head. For a sample $(X, Y)$, we define $\hat{v} = h(X)$ as the trained feature. We follow the ideas in Teng et al. [2022] and define the surrogate feature as any feature $v$ such that $g(v) = Y$.

**Assumption 1** (exchangeability). *Assume that the calibration data $(X_i, Y_i) \in \mathcal{D}_{cal}$ and the testing point $(X', Y')$ are exchangeable. Formally, define $Z_i, i = 1, \ldots, |\mathcal{I}_{cal}| + 1$, as the above data pair. Then $Z_i$ are exchangeable if arbitrary permutation follows the same distribution, i.e.,*

$$(Z_1, \ldots, Z_{|\mathcal{I}_{cal}|+1}) \stackrel{d}{=} (Z_{\pi(1)}, \ldots, Z_{\pi(|\mathcal{I}_{cal}|+1)}), \tag{3}$$

*with arbitrary permutation $\pi$ over $\{1, \cdots, |\mathcal{I}_{cal}| + 1\}$.*

Typically, Split CP is composed of three key steps.
**I. Training Step.** We first train a base model using the training fold $\mathcal{D}_{\text{tra}}$.
**II. Calibration Step.** We calculate a non-conformity score $R_i = |Y_i - f(X_i)|$ using the calibration fold $\mathcal{D}_{\text{cal}}$. The form of the score function might vary case by case, quantifying the divergence between ground truth and predicted values.
**III. Testing Step.** We construct the confidence band for the testing point $(X', Y')$ using the quantile of the non-conformity score $Q_{1-\alpha}$.

We present Split CP* in Algorithm 1, and provide its theoretical guarantee in Theorem 3.1.

**Theorem 3.1.** *Under Assumption 1, the confidence band $\mathcal{C}_{1-\alpha}(X')$ returned by Algorithm 1 satisfies*

$$\mathbb{P}(Y' \in \mathcal{C}_{1-\alpha}(X')) \geq 1 - \alpha. \tag{4}$$

## 4 Methodology

In this section, we first illustrate the motivation behind FFCP in Section 4.1. Specifically, we address the complexity of non-linear operators in FCP and derive FFCP from FCP. We then formally present the specific form of FFCP, including the non-conformity score, the returned bands, and the corresponding pseudocode. We finally provide theoretical analyses on the coverage and band length in Section 4.2.

---

*$\delta$ represents the Dirac function.

---

**Algorithm 1** Split Conformal Prediction

---

**Input:** Confidence level $\alpha$, dataset $\mathcal{D} = \{(X_i, Y_i)\}_{i \in \mathcal{I}}$, testing point $X'$

1: Randomly split the dataset $\mathcal{D}$ into a training fold $\mathcal{D}_{\text{tra}} \triangleq \{(X_i, Y_i)\}_{i \in \mathcal{I}_{\text{tra}}}$ and a calibration fold $\mathcal{D}_{\text{cal}} \triangleq \{(X_i, Y_i)\}_{i \in \mathcal{I}_{\text{cal}}}$;
2: Train a base model $f(\cdot)$ with training fold $\mathcal{D}_{\text{tra}}$ ;
3: For each $i \in \mathcal{I}_{\text{cal}}$, calculate the non-conformity score $R_i = |Y_i - f(X_i)|$;
4: Calculate the $(1-\alpha)$-th quantile $Q_{1-\alpha}$ of the distribution $\frac{1}{|\mathcal{I}_{\text{cal}}|+1} \sum_{i \in \mathcal{I}_{\text{cal}}} \delta_{R_i} + \delta_\infty$.

**Output:** $\mathcal{C}_{1-\alpha}^{\text{Splitcp}}(X') = [f(X') - Q_{1-\alpha}, f(X') + Q_{1-\alpha}]$.

---

---

**Algorithm 2** Fast Feature Conformal Prediction

---

**Input:** Confidence level $\alpha$, dataset $\mathcal{D} = \{(X_i, Y_i)\}_{i \in \mathcal{I}}$, testing point $X'$

1: Randomly split the dataset $\mathcal{D}$ into a training fold $\mathcal{D}_{\text{tra}} \triangleq \{(X_i, Y_i)\}_{i \in \mathcal{I}_{\text{tra}}}$ and a calibration fold $\mathcal{D}_{\text{cal}} \triangleq \{(X_i, Y_i)\}_{i \in \mathcal{I}_{\text{cal}}}$ ;
2: Train a base neural network with training fold $f(\cdot) = g \circ h(\cdot)$ with training fold $\mathcal{D}_{\text{tra}}$;
3: For each $i \in \mathcal{I}_{\text{cal}}$, calculate the non-conformity score $\tilde{R}_i = |Y_i - f(X_i)|/\|\nabla g(\hat{v}_i)\|$, where $\nabla g(\hat{v}_i)$ denotes the gradient of $g(\cdot)$ on the feature $\hat{v}_i \triangleq h(X_i)$, namely $\nabla g(\hat{v}_i) = \frac{\mathrm{d} g \circ h(X_i)}{\mathrm{d} h(X_i)}$;
4: Calculate the $(1-\alpha)$-th quantile $Q_{1-\alpha}$ of the distribution $\frac{1}{|\mathcal{I}_{\text{cal}}|+1} \sum_{i \in \mathcal{I}_{\text{cal}}} \delta_{\tilde{R}_i} + \delta_\infty$;

**Output:** $\mathcal{C}_{1-\alpha}^{\text{ffcp}}(X') = [f(X') - \|\nabla g(\hat{v}')\|Q_{1-\alpha}, \ f(X') + \|\nabla g(\hat{v}')\|Q_{1-\alpha}]$, where $\hat{v}' = h(X')$.

---

## 4.1 Relationship between FFCP and FCP

In this section, we discuss the motivation behind FFCP. FFCP is inspired by FCP [Teng et al., 2022], which conducts conformal prediction in the feature space. However, since the band is constructed in the feature space, FCP requires a *Band Estimation* process to go from feature space to output space. Specifically, FCP applies *LiPRA* [Xu et al., 2020] which derives the band in the output space $\{g(v) : \|v - \hat{v}\| \leq Q_{1-\alpha}\}$. Unfortunately, the exact band is difficult to represent explicitly since the prediction head $g$ is usually highly non-linear, thereby resulting in significant computational complexity in terms of time. Therefore, we propose approximating $g$ using a first-order Taylor expansion to simplify the aforementioned non-linear operator. The core steps of FCP include (a) calculating the non-conformity score (from output space to feature space), followed by (b) deriving the confidence band (from feature space to output space). We next introduce the concrete formulation of how FFCP approximates FCP.

**From output space to feature space.** FCP uses the non-conformity score $s_{\text{f}}(\cdot)$ in the feature space:

$$s_{\text{f}}(X, Y, g \circ h) = \inf_{v \in \{v : g(v) = Y\}} \|v - \hat{v}\|. \tag{5}$$

By using the Taylor expansion, one approximates $g$ with $g(v) \approx g(\hat{v}) + \nabla g(\hat{v})(v - \hat{v})$. Plugging into the approximation of $g$ leads to a new non-conformity score $s_{\text{ff}}(\cdot)$

$$s_{\text{ff}}(X, Y, g \circ h) = |Y - f(X)|/\|\nabla g(\hat{v})\|, \tag{6}$$

where $\nabla g(\hat{v})$ denotes the gradient of $g(\hat{v})$ on the feature $\hat{v}$, namely $\nabla g(\hat{v}) = \frac{\mathrm{d} g \circ h(X)}{\mathrm{d} h(X)}$.

**From feature space to output space.** After constructing the confidence band in the feature space, FCP maps this band to the output space. Specifically, FCP derives the following band in the output space which is called *Band Estimation*:

$$\{g(v) : \|v - \hat{v}\| \leq Q_{1-\alpha}\}. \tag{7}$$

FCP proposes to use LiPRA in this process, which is time-consuming. By plugging into the Taylor approximation of $g$, one can construct the band $\mathcal{C}_{1-\alpha}^{\text{ffcp}}$ as

$$\mathcal{C}_{1-\alpha}^{\text{ffcp}}(X) = [g(\hat{v}) - \|\nabla g(\hat{v})\|Q_{1-\alpha}, g(\hat{v}) + \|\nabla g(\hat{v})\|Q_{1-\alpha}]. \tag{8}$$

**Remark 1** (High-dimensional Response)**.** *When the response* $Y_i = [Y_i^1, Y_i^2, \ldots, Y_i^m], i = 1, 2, \cdots, n$ *is high-dimensional, one can deploy conformal prediction at a coordinate-wise level.*

*Specifically, for dimension $j \in [m]$, we define the non-conformity score as*

$$s_{ff}^j(X_i, Y_i, g \circ h) = |Y_i^j - f(X_i)^j| / \|\nabla g(\hat{v}_i)^j\|, \tag{9}$$

*where $\nabla g(\hat{v}_i)^j = \left( \frac{\partial f(X_i)}{\partial h(X_i)} \right)_j$ represents the $j$-th row of the Jacobian matrix of $f$ with respect to $h$ at $X_i$.*

*We then compute a single quantile $Q_{1-\alpha}$ shared across all dimensions, defined by aggregating non-conformity scores from all coordinates and samples in the calibration set:*

$$Q_{1-\alpha} = Quantile\left( \left\{ s_{ff}^j(X_i, Y_i, g \circ h) : i \in \mathcal{I}_{cal}, j \in [m] \right\}, 1 - \alpha \right). \tag{10}$$

*The resulting confidence band for the $j$-th coordinate at test point $X_i$ is given by:*

$$\mathcal{C}_{1-\alpha}^{ffcp}(X_i)_j = \left[ g(\hat{v}_i)^j - \|\nabla g(\hat{v}_i)^j\| Q_{1-\alpha}, \ g(\hat{v}_i)^j + \|\nabla g(\hat{v}_i)^j\| Q_{1-\alpha} \right]. \tag{11}$$

Based on the above discussion, we present the full algorithm in Algorithm 2. Notably, the Taylor expansion in FFCP is usually different for each sample $X, Y$, which further leads to confidence bands that are individually different. Besides, FFCP inherits the advantages of FCP. For example, this framework is pretty general and can be combined with other variants of Split CP, *e.g.*, CQR [Romano et al., 2019a].

## 4.2 Theoretical Guarantee for FFCP

This section outlines the theoretical guarantee for FFCP concerning coverage (effectiveness) and band length (efficiency). Below, we offer the main theorems and defer the full proofs to Appendix A.1 and A.2. We first demonstrate that the confidence band produced by Algorithm 2 is valid under Assumption 1.

**Theorem 4.1** (Coverage)**.** *Under Assumption 1, for any $\alpha > 0$, the confidence band returned by Algorithm 2 satisfies:*

$$\mathbb{P}(Y' \in \mathcal{C}_{1-\alpha}^{ffcp}(X')) \geq 1 - \alpha, \tag{12}$$

*where the probability is taken over the calibration fold and the testing point $(X', Y')$.*

Next, we show that FFCP is provably more efficient than the Split CP. To simplify the discussion, we present an informal version of Theorem 4.2 here and postpone the formal version to Theorem A.1.

**Theorem 4.2** (Band Length)**.** *Under mild assumptions, if the following square conditions hold:*

1. ***Expansion.** The feature space expands the differences between individual lengths and their quantiles.*

2. ***Quantile Stability.** Given a calibration set $\mathcal{D}_{cal}$, the quantile of the band length is stable in both feature space and output space.*

*Then FFCP provably outperforms Split CP in terms of average band length.*

By Theorem 4.2, FFCP is guaranteed to achieve a smaller average band length than Split CP. The *square conditions* imply that the feature space has a smaller distance between individual non-conformity scores and their quantiles. This reduction in the computational overhead of the quantile operation subsequently yields a shorter band length. We provide empirical verifications on this assumption, see Figure 4 for more details. The intuition behind Theorem 4.2 is as follows: Initially, FFCP and Split CP perform quantile operations in different spaces, with the *Expansion* condition ensuring that the quantile step in FFCP costs less. The ultimate *Quantile Stability* condition confirms that the band can be generalized from the calibration folds to the test folds.

## 5 Experiments

This section presents the experiments to validate the utility of FFCP. Firstly, we detail the experimental setup in Section 5.1. Secondly, we present that FFCP achieves both effectiveness and efficiency with

Table 1: Time comparison among Split CP, FCP, and FFCP. FFCP ensures faster running speed compared to FCP. The last column represents the speed improvement factor of FFCP compared to FCP. The time unit is in seconds.

| DATASET | SPLIT CP | FCP | FFCP | FASTER |
|---|---|---|---|---|
| SYNTHETIC | $0.0088\pm0.0003$ | $3.8939\pm0.3725$ | $0.0902\pm0.0056$ | 43x |
| COM | $0.0047\pm0.0010$ | $4.9804\pm0.8588$ | $0.0844\pm0.0187$ | 59x |
| FB1 | $0.0245\pm0.0059$ | $5.9822\pm0.9871$ | $0.1940\pm0.0564$ | 31x |
| FB2 | $0.0414\pm0.0070$ | $9.3534\pm0.0927$ | $0.2510\pm0.0058$ | 37x |
| MEPS19 | $0.0106\pm0.0010$ | $3.3237\pm0.0431$ | $0.0755\pm0.0037$ | 44x |
| MEPS20 | $0.0152\pm0.0016$ | $5.4003\pm0.3945$ | $0.0948\pm0.0077$ | 57x |
| MEPS21 | $0.0137\pm0.0008$ | $4.1657\pm0.0670$ | $0.0854\pm0.0146$ | 49x |
| STAR | $0.0030\pm0.0006$ | $3.5842\pm0.3722$ | $0.0332\pm0.0066$ | 108x |
| BIO | $0.0291\pm0.0053$ | $7.5417\pm1.1028$ | $0.2042\pm0.0344$ | 37x |
| BLOG | $0.0340\pm0.0024$ | $8.0913\pm1.2072$ | $0.2239\pm0.0261$ | 36x |
| BIKE | $0.0072\pm0.0007$ | $3.5806\pm0.0285$ | $0.0534\pm0.0021$ | 67x |

Table 2: Comparisons of coverage and band length among Split CP, FCP, and FFCP. FFCP runs faster while performing comparably to FCP in most datasets and outperforming Split CP. For FFCP, we select the shortest band length among all layers.

| METHOD | SPLIT CP | | FCP | | FFCP | |
|---|---|---|---|---|---|---|
| DATASET | COVERAGE | LENGTH | COVERAGE | LENGTH | COVERAGE | LENGTH |
| SYNTHETIC | $90.080\pm0.951$ | $0.176\pm0.015$ | $89.930\pm0.956$ | $\mathbf{0.081}\pm0.041$ | $90.080\pm0.951$ | $0.176\pm0.015$ |
| COM | $89.875\pm0.985$ | $1.974\pm0.071$ | $89.724\pm1.087$ | $1.939\pm1.408$ | $90.226\pm2.179$ | $\mathbf{1.838}\pm0.180$ |
| FB1 | $90.254\pm0.170$ | $2.004\pm0.191$ | $90.198\pm0.207$ | $2.010\pm0.182$ | $90.168\pm0.220$ | $\mathbf{1.472}\pm0.232$ |
| FB2 | $89.933\pm0.206$ | $2.016\pm0.218$ | $89.966\pm0.130$ | $\mathbf{1.371}\pm0.370$ | $89.868\pm0.062$ | $1.425\pm0.109$ |
| MEPS19 | $90.567\pm0.311$ | $3.982\pm0.614$ | $90.605\pm0.340$ | $3.493\pm2.734$ | $90.352\pm0.469$ | $\mathbf{3.134}\pm0.309$ |
| MEPS20 | $89.923\pm0.715$ | $4.184\pm0.316$ | $89.929\pm0.770$ | $\mathbf{2.730}\pm0.962$ | $89.615\pm0.661$ | $3.268\pm0.283$ |
| MEPS21 | $90.019\pm0.341$ | $3.732\pm0.555$ | $90.038\pm0.303$ | $3.393\pm1.313$ | $89.745\pm0.344$ | $\mathbf{3.146}\pm0.506$ |
| STAR | $90.393\pm1.494$ | $0.208\pm0.004$ | $90.300\pm1.362$ | $\mathbf{0.174}\pm0.038$ | $90.393\pm1.494$ | $0.208\pm0.004$ |
| BIO | $89.875\pm0.488$ | $1.661\pm0.019$ | $89.930\pm0.501$ | $\mathbf{1.412}\pm0.265$ | $89.875\pm0.488$ | $1.661\pm0.019$ |
| BLOG | $90.176\pm0.241$ | $3.524\pm0.850$ | $90.151\pm0.405$ | $2.795\pm1.385$ | $90.059\pm0.101$ | $\mathbf{2.741}\pm0.517$ |
| BIKE | $89.871\pm0.568$ | $0.703\pm0.016$ | $89.394\pm0.633$ | $2.147\pm0.249$ | $89.624\pm0.688$ | $\mathbf{0.635}\pm0.030$ |

faster execution in Section 5.2. Thirdly, in Section 5.3.1, we verify that FFCP can be easily deployed and performs robustly across various tasks, including classification and segmentation. Finally, in Section 5.3.2, we show that the gradient-level techniques used in FFCP can be extended to classic CP models such as CQR [Romano et al., 2019a] and LCP [Guan, 2023]. A more detailed account of this extension can be found in Section 5.3.

### 5.1 Experiments Setups

**Datasets.** We consider both synthetic datasets and realistic datasets, including **(a) synthetic dataset**: $Y = WX + \epsilon$, where $X \in [0,1]^{100}, Y \in \mathbb{R}, \epsilon \sim \mathcal{N}(0,1)$, $W$ is a fixed random matrix. **(b) real-world unidimensional target datasets**: ten datasets from UCI machine learning [Asuncion, 2007] and other sources: community and crimes (*COM*), Facebook comment volume variants one and two (*FB1* and *FB2*), medical expenditure panel survey (*MEPS19–21*) [Cohen et al., 2009], Tennessee's student teacher achievement ratio (*STAR*) [Achilles et al., 2008], physicochemical properties of protein tertiary structure (*BIO*), blog feedback (*BLOG*) [Buza, 2014], and bike sharing (*BIKE*), **(c) real-world semantic segmentation dataset**: Cityscapes [Cordts et al., 2016], and **(d) real-world semantic classification dataset**: Imagenet-Val [Deng et al., 2009].

**Algorithms.** We compare three methods: Split CP, FCP, and FFCP, with Split CP serving as the baseline. For the one-dimensional scenario, we perform direct calculations. For higher-dimensional cases, we use a coordinate-wise level non-conformity score.

**Evaluation.** The algorithmic empirical performance is evaluated with the following metrics:

Table 3: Coverage and Band Length based on Gradient from Different Layers of Neural Networks. FFCP LAYER(·) represents using the gradient between the LAYER(·) and the input. The results in LAYER4 are equivalent to Split CP.

| LAYER | LAYER1 | | LAYER2 | | LAYER3 | | LAYER4 | |
|---|---|---|---|---|---|---|---|---|
| DATASET | COVERAGE | LENGTH | COVERAGE | LENGTH | COVERAGE | LENGTH | COVERAGE | LENGTH |
| SYNTHETIC | $89.810 \pm 0.784$ | $0.184 \pm 0.018$ | $90.050 \pm 0.534$ | $0.184 \pm 0.017$ | $89.960 \pm 0.910$ | $\mathbf{0.182} \pm 0.023$ | $90.220 \pm 0.983$ | $0.189 \pm 0.033$ |
| COM | $90.476 \pm 1.889$ | $1.878 \pm 0.224$ | $90.226 \pm 2.179$ | $\mathbf{1.838} \pm 0.180$ | $89.674 \pm 1.465$ | $1.853 \pm 0.136$ | $89.825 \pm 0.646$ | $2.037 \pm 0.188$ |
| FB1 | $90.112 \pm 0.199$ | $3.540 \pm 0.327$ | $90.212 \pm 0.357$ | $2.860 \pm 0.327$ | $90.083 \pm 0.216$ | $1.597 \pm 0.052$ | $90.168 \pm 0.220$ | $\mathbf{1.472} \pm 0.232$ |
| FB2 | $89.953 \pm 0.250$ | $3.530 \pm 0.384$ | $89.897 \pm 0.235$ | $3.048 \pm 0.510$ | $89.956 \pm 0.159$ | $2.077 \pm 0.517$ | $89.868 \pm 0.062$ | $\mathbf{1.425} \pm 0.109$ |
| MEPS19 | $90.155 \pm 0.643$ | $3.251 \pm 0.396$ | $90.352 \pm 0.469$ | $\mathbf{3.134} \pm 0.309$ | $90.440 \pm 0.183$ | $3.184 \pm 0.482$ | $90.586 \pm 0.246$ | $3.795 \pm 0.640$ |
| MEPS20 | $89.934 \pm 0.520$ | $4.302 \pm 1.377$ | $89.889 \pm 0.621$ | $3.573 \pm 0.488$ | $89.615 \pm 0.661$ | $\mathbf{3.268} \pm 0.283$ | $89.82 \pm 0.689$ | $3.817 \pm 0.308$ |
| MEPS21 | $89.496 \pm 0.262$ | $3.443 \pm 0.487$ | $89.623 \pm 0.275$ | $3.218 \pm 0.239$ | $89.745 \pm 0.344$ | $\mathbf{3.146} \pm 0.506$ | $90.026 \pm 0.301$ | $3.452 \pm 0.711$ |
| STAR | $90.901 \pm 1.732$ | $0.221 \pm 0.002$ | $90.993 \pm 1.807$ | $0.217 \pm 0.003$ | $91.039 \pm 1.442$ | $0.210 \pm 0.004$ | $90.300 \pm 1.248$ | $\mathbf{0.209} \pm 0.004$ |
| BIO | $89.937 \pm 0.391$ | $2.292 \pm 0.077$ | $90.022 \pm 0.375$ | $2.042 \pm 0.067$ | $89.991 \pm 0.594$ | $2.080 \pm 0.063$ | $90.127 \pm 0.476$ | $\mathbf{1.822} \pm 0.025$ |
| BLOG | $89.968 \pm 0.420$ | $4.772 \pm 0.614$ | $89.918 \pm 0.319$ | $3.404 \pm 0.598$ | $90.059 \pm 0.101$ | $\mathbf{2.741} \pm 0.517$ | $90.017 \pm 0.197$ | $3.058 \pm 0.873$ |
| BIKE | $89.917 \pm 0.791$ | $1.701 \pm 0.254$ | $89.568 \pm 0.476$ | $1.138 \pm 0.114$ | $89.495 \pm 0.579$ | $0.794 \pm 0.068$ | $89.624 \pm 0.688$ | $\mathbf{0.635} \pm 0.030$ |

- **Runtime** For runtime evaluation, the timing starts at the score calculation and ends with the final prediction bands returned. FFCP method records the total computation time for each layer, and then selects the layer that achieves the best results.

- **Coverage (Effectiveness)** Coverage refers to the observed frequency with which a test point falls within the predicted confidence interval. Ideally, a predictive inference method should yield a coverage rate slightly higher than $1 - \alpha$ for a given significance level $\alpha$.

- **Band length (Efficiency)** When the coverage exceeds $1 - \alpha$, our goal is to minimize the length of the confidence band. For FFCP, since we use a 5-layers neural network, each layer can be viewed as a feature layer. Therefore, in the experiments, we obtain the band length returned by each of the 5 layers of the neural network. In the subsequent results, if only a single band length is presented, it corresponds to the shortest band length returned by the different neural network layers. Otherwise, the results for all layers from layer 0 to layer 4 (with the last layer typically representing the Split CP result) will be shown.

Let $Y = (Y^1, \ldots, Y^d) \in \mathbb{R}^d$ denote the $d$-dimensional response variable, and let $\mathcal{C}(X) \subseteq \mathbb{R}^d$ be the confidence band associated with predictor $X$. The length of this confidence band in each dimension is represented by the vector $(|\mathcal{C}(X)^1|, \ldots, |\mathcal{C}(X)^d|) \in \mathbb{R}^d$. Denote the indices of the test set by $\mathcal{I}_{\text{tes}}$ and the set of dimensions by $[d] = \{1, \ldots, d\}$. We then define the coverage and band length as:

$$\text{Coverage} = \frac{1}{|\mathcal{I}_{\text{tes}}|} \sum_{i \in \mathcal{I}_{\text{tes}}} \mathbb{I}\left(Y_i \in \mathcal{C}(X_i)\right), \quad \text{Band Length} = \frac{1}{|\mathcal{I}_{\text{tes}}|} \sum_{i \in \mathcal{I}_{\text{tes}}} \left(\frac{1}{d} \sum_{j=1}^{d} |\mathcal{C}(X_i)^j|\right), \quad (13)$$

where $\mathbb{I}(\cdot)$ is the indicator function that equals 1 if its argument is true and 0 otherwise.

## 5.2 Results on Coverage, Band Length and Runtime

**Runtime Comparison.** The runtime comparison is presented in Table 1. The results show that FFCP outperforms FCP with an approximate 50x speedup in runtime. Notably, since Split CP is the most basic method and does not utilize additional tools, it exhibits the fastest runtime.

**Coverage.** Table 2 summarizes the coverage for the one-dimensional response. Experimental results indicate that the coverage of FFCP all exceeds the confidence level $1 - \alpha$, affirming its effectiveness as stated in Theorem 4.1.

**Band Length.** The band length is detailed in Table 2 for a one-dimensional response. It is noteworthy that FFCP surpasses Split CP by achieving a shorter band length, thereby validating the efficiency of the algorithm.

## 5.3 Extensions of FFCP

This section provides the extensions of FFCP, which is divided into two parts. Section 5.3.1 mainly discusses the applications of FFCP beyond regression tasks, specifically in image classification [Angelopoulos et al., 2020] and segmentation tasks. Section 5.3.2 focuses on how the gradient-level

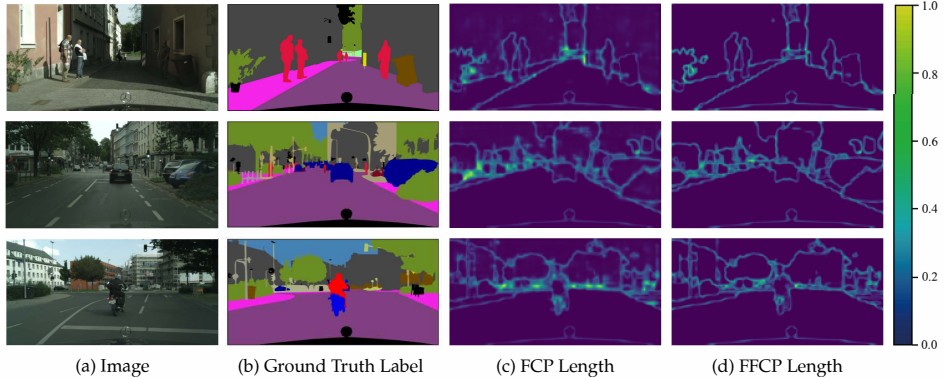

| (a) Image | (b) Ground Truth Label | (c) FCP Length | (d) FFCP Length |

Figure 2: Segmentation uncertainty (FFCP vs. FCP). Both FFCP and FCP capture uncertainty concentrated around object boundaries, with FFCP producing more refined and sharper uncertainty bands. Notably, the two methods aim at slightly different goals: FCP operates at the image level, ensuring coverage for the entire predicted mask as a whole, while FFCP adopts a coordinate-wise approach, offering per-pixel statistical guarantees that better reflect local uncertainty.

techniques in FFCP can be extended to other CP variants, *e.g.*, CQR [Romano et al., 2019a] and LCP [Guan, 2023].

### 5.3.1 Other Tasks

**Classification.** We extend the FFCP techniques to classification tasks using the baseline RAPS [Angelopoulos et al., 2020] model, creating a new variant called FFRAPS (Fast Feature RAPS, Algorithm 5 in Appendix B.7). According to the experimental findings presented in Table 15, FFRAPS returns shorter band lengths while preserving the coverage compared to RAPS under most model structures.

**Segmentation.** The gradient-level techniques of FFCP also prove effective in segmentation tasks. The segmentation results in Figure 2 reveal that FFCP returns appropriate bands across different regions. Specifically, larger bands are observed in less informative areas, such as at object boundaries, whereas narrower bands are found in more informative regions. This validates the efficiency of FFCP in segmentation tasks.

### 5.3.2 Extending FFCP into Other Models

**Conformalized Quantile Regression (CQR, Romano et al. [2019a])** The gradient-level techniques of FFCP are adaptable to other conformal prediction frameworks like CQR. We develop FFCQR (Fast Feature CQR, Algorithm 3 in Appendix B.5), which not only significantly reduces runtime compared to FCQR but also exhibits better performance than CQR. Additionally, we observe that for the neural network significant level setting $[\alpha, 1 - \alpha]$ in the CQR method, as the $\alpha$ value increases, approaching $1 - \alpha/2$, the performance of FFCQR gradually improves. For detailed experimental results in Table 8,9, 10, 11, 12 in the Appendix B.5.

**Locally Adaptive Conformal Prediction (LCP, Guan [2023])** Integrating gradient-level techniques from FFCP into the LCP method leads to FFLCP (Fast Feature LCP, Algorithm 4 in Appendix B.6). Experimental results in Table 13 indicate that FFLCP outperforms LCP in terms of group coverage, highlighting an improvement in the adaptability of LCP to locally adaptive methods.

### 5.3.3 Comparison of FFCP with Other Baselines

**Self-Supervised Conformal Prediction (SSCP, Seedat et al. [2023])** We additionally evaluate the recent feature-CP method SSCP in Appendix B.10. SSCP entails training two extra networks and is roughly 50× slower than FFCP. In all cases, FFCP also yields shorter bands, likely because SSCP depends more heavily on the base model's accuracy and on effective auxiliary-network training.

**Full CP with CV+ [Barber et al., 2021] framework**  Although FFCP is a Split CP procedure, we also benchmark it against the full CP baseline CV+ and observe consistent advantages—most notably improved efficiency with comparable coverage. Complete experimental details and results are provided in the Appendix B.9.

# 6  Conclusion

In this paper, we propose FFCP, a gradient-based non-conformity score that is 50× faster than FCP. We establish its theoretical validity under mild assumptions and demonstrate its broad applicability across regression, classification, and segmentation tasks. We also introduce FFCQR and FFLCP, based on CQR and LCP, respectively. Finally, we evaluate FFCP in comparison with SSCP and the full CP baseline, CV+.

Although FFCP is gradient-based, it can be extended to settings where gradients are not available or are unreliable—*e.g.*, when gradients vanish or the function is non-differentiable—by incorporating zero-th order methods such as finite difference approximations or perturbation-based surrogates. This further broadens the applicability of our framework beyond differentiable models. For future work, the following points could be considered: (1) We use information from the first derivative and have not delved into higher-order derivatives, which may contain more feature information; (2) The gradient at a single point may be unstable, especially when the gradient is zero, so methods such as random smoothing could be considered.

**Acknowledgments**

This work is supported by Shanghai Science and Technology Development Funds 24YF2711700 and Fundamental Research Funds for the Central Universities 2024110586. The authors would like to express their gratitude to Dinghuai Zhang for his constructive insights and valuable discussions.

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

# Appendix

The complete proofs are presented in Section A, and the experiment details are outlined in Section B.

## A  Theoretical Proofs

We prove the theoretical guarantee for FFCP concerning coverage (effectiveness) in Section A.1 and band length (efficiency) in Section A.2.

### A.1  Proofs of Theorem 4.1

The proof is based on the exchangeability of data (Assumption 1) on the calibration fold and test fold, hence the key step we need to derive is the exchangeability of the non-conformity scores $s_{\text{ff}}(X, Y, g \circ h) = |Y - f(X)|/\|\nabla g(\hat{v})\|$. We define the relevant symbols: $\mathcal{D}_{\text{tra}}$ represents the train fold, $\mathcal{D}_{\text{tes}}$ represents the test fold, $\mathcal{D}_{\text{cal}}$ represents the calibration fold, and $\mathcal{D}' = \{(X_i, Y_i)\}_{i \in [m]}$ is the intersection of the two folds. $m$ is the number of data points in $\mathcal{D}'$.

Similar to Teng et al. [2022], we first prove that for any function $\tilde{h} : \mathcal{X} \times \mathcal{Y} \to \mathbb{R}$, which is independent of $\mathcal{D}'$, $\tilde{h}(X_i, Y_i)$ satisfies exchangeability. For the CDF $F_R$ of $\tilde{h}$ and its perturbation CDF $F_R^\pi$, $\pi$ is a random perturbation. We can conclude,

$$
\begin{aligned}
&F_R(u_1, \ldots, u_n \mid \mathcal{D}_{\text{tra}}) \\
=&\mathbb{P}(\tilde{h}(X_1, Y_1) \le u_1, \ldots, \tilde{h}(X_n, Y_n) \le u_n \mid \mathcal{D}_{\text{tra}}), \\
=&\mathbb{P}((X_1, Y_1) \in \mathcal{C}_{\tilde{h}^{-1}}(u_1-), \ldots, (X_n, Y_n) \in \mathcal{C}_{\tilde{h}^{-1}}(u_n-) \mid \mathcal{D}_{\text{tra}}), \\
=&\mathbb{P}((X_{\pi(1)}, Y_{\pi(1)}) \in \mathcal{C}_{\tilde{h}^{-1}}(u_1-), \ldots, (X_{\pi(n)}, Y_{\pi(n)}) \in \mathcal{C}_{\tilde{h}^{-1}}(u_n-) \mid \mathcal{D}_{\text{tra}}), \\
=&\mathbb{P}(\tilde{h}(X_{\pi(1)}, Y_{\pi(1)}) \le u_1, \ldots, \tilde{h}(X_{\pi(n)}, Y_{\pi(n)}) \le u_n \mid \mathcal{D}_{\text{tra}}), \\
=&F_R^\pi(u_1, \ldots, u_n \mid \mathcal{D}_{\text{tra}}),
\end{aligned}
\tag{14}
$$

where $\mathcal{C}_{\tilde{h}^{-1}}(u-) = \{(X, Y) : \tilde{h}(X, Y) \le u\}$.

Next, we need to show the non-conformity score function

$$
s_{\text{ff}}(X, Y, g \circ h) = |Y - f(X)|/\|\nabla g(\hat{v})\|,
\tag{15}
$$

which is independent of the dataset $\mathcal{D}'$.

We can see that the non-conformity score $s_{\text{ff}}(X, Y, g \circ h)$ on $\mathcal{D}'$ uses information from $g$ and $h$, both of which depend only on the training set $\mathcal{D}_{\text{tra}}$. Moreover, calculating this non-conformity score in the Algorithm 2 uses only single-point information, not the entire dataset $\mathcal{D}'$.

By integrating the aforementioned, we deduce that the non-conformity scores $s_{\text{ff}}(X, Y, g \circ h)$ on $\mathcal{D}'$ exhibit exchangeability. This exchangeability, as per Lemma 1 in Tibshirani et al. [2019], lends theoretical support to the efficacy of FFCP.

### A.2  Proofs of Theorem 4.2

Our main conclusions are inspired by Theorem 4 in Teng et al. [2022]. The details are as follows

**Definitions.** Let $\mathcal{P}$ denote the overall population distribution. The calibration set $\mathcal{D}_{\text{cal}}$ consists of $n$ samples drawn from $\mathcal{P}$. We denote the specific distribution of these samples as $\mathcal{P}^n$. The model under consideration, $f = g \circ h$, includes $h$ as the feature extractor and $g$ as the prediction head, with $g$ assumed to be a continuous function. $V_{\mathcal{D}}^o$ represent the individual length in output space, given data set $\mathcal{D}$. The term $Q_{1-\alpha}(R)$ represents the $(1 - \alpha)$-quantile of the set $R$, which is adjusted to include the value 0. Furthermore, $\mathbb{M}[\cdot]$ signifies the mean value of a set, and subtracting a real number from a set indicates that the subtraction is applied uniformly to all elements within the set.

*Split CP.* Let $V_{\mathcal{D}_{\text{cal}}}^o = \{v_i^o\}_{i \in \mathcal{I}_{\text{cal}}}$ denote the individual length in the output space for Split CP, given the calibration set $\mathcal{D}_{\text{cal}}$. Since Split CP returns band length with $1 - \alpha$ quantile of non-conformity score, the resulting average band length is derived by $2Q_{1-\alpha}(V_{\mathcal{D}_{\text{cal}}}^o)$.

*Fast Feature CP.* According to the definition of FFCP, $V_{\mathcal{D}}^{f} = V_{\mathcal{D}}^{o}/\|\nabla g(\hat{v})\|$,

The resulting band length in FFCP is denoted by $2\mathbb{E}_{(X',Y')\sim\mathcal{P}}(\|\nabla g(\hat{v'})\| \cdot Q_{1-\alpha}(V_{\mathcal{D}_{\mathrm{cal}}}^{o}/\|\nabla g(\hat{v}_{\mathrm{cal}})\|))$.

**Theorem A.1.** *(FFCP is provably more efficient). Assume that the non-conformity score is in norm-type. We assume a Holder assumption that there exist $\alpha > 0, L > 0$ such that $|\mathcal{H}(x; X) - \mathcal{H}(y; X)| \leq L|x - y|^{\alpha}$ for all $X$, where $\mathcal{H}$ denotes the length of the prediction interval in the output space for samples. Then if the feature space satisfies the following square conditions:*

1. ***Expansion.*** *The feature space expands the differences between individual length and their quantiles, namely, $L\mathbb{E}_{\mathcal{D}\sim\mathcal{P}^{n}}\mathbb{M}|Q_{1-\alpha}(V_{\mathcal{D}}^{o}/\|\nabla g(\hat{v})\|) - V_{\mathcal{D}}^{o}/\|\nabla g(\hat{v})\||^{\alpha} < \mathbb{E}_{\mathcal{D}\sim\mathcal{P}^{n}}\mathbb{M}[Q_{1-\alpha}(V_{\mathcal{D}}^{o}) - V_{\mathcal{D}}^{o}] - 2\max\{L, 1\}(c/\sqrt{n})^{\min\{\alpha, 1\}}$.*

2. ***Quantile Stability.*** *Given a calibration set $\mathcal{D}_{cal}$, the quantile of the band length is stable in both feature space and output space, namely, $\mathbb{E}_{\mathcal{D}\sim\mathcal{P}^{n}}|Q_{1-\alpha}(V_{\mathcal{D}}^{o}/\|\nabla g(\hat{v})\|) - Q_{1-\alpha}(V_{\mathcal{D}_{cal}}^{o}/\|g(\nabla\hat{v}_{cal})\|)| \leq \frac{c}{\sqrt{n}}$ and $\mathbb{E}_{\mathcal{D}\sim\mathcal{P}^{n}}|Q_{1-\alpha}(V_{\mathcal{D}}^{o}) - Q_{1-\alpha}(V_{\mathcal{D}_{cal}}^{o})| \leq \frac{c}{\sqrt{n}}$.*

*Then FFCP provably outperforms Split CP in terms of average band length, namely,*

$$\mathbb{E}_{(X',Y')\sim\mathcal{P}}(\|\nabla g(\hat{v'})\| \cdot Q_{1-\alpha}(V_{\mathcal{D}_{cal}}^{o}/\|\nabla g(\hat{v}_{cal})\|)) < Q_{1-\alpha}(V_{D_{cal}}^{0}),$$

*where the expectation is taken over the calibration fold and the testing point $(X', Y')$.*

*Proof of Theorem A.1.* We first proof with *Expansion* Assumption,

$$\begin{aligned}L\mathbb{E}_{\mathcal{D}\sim\mathcal{P}^{n}}\mathbb{M}|Q_{1-\alpha}(V_{\mathcal{D}}^{o}/\|\nabla g(\hat{v})\|) - V_{\mathcal{D}}^{o}/\|\nabla g(\hat{v})\||^{\alpha} &< \mathbb{E}_{\mathcal{D}\sim\mathcal{P}^{n}}\mathbb{M}[Q_{1-\alpha}(V_{\mathcal{D}}^{o}) - V_{\mathcal{D}}^{o}] \\ &- 2\max\{L, 1\}(c/\sqrt{n})^{\min\{\alpha, 1\}}.\end{aligned} \tag{16}$$

And we can obtain

$$\begin{aligned}\mathbb{E}_{\mathcal{D}}\mathbb{M}V_{\mathcal{D}}^{o} < &\mathbb{E}_{\mathcal{D}}Q_{1-\alpha}(V_{\mathcal{D}}^{o}) \\ &- 2\max\{L, 1\}(c/\sqrt{n})^{\min\{\alpha, 1\}} - L\mathbb{E}_{\mathcal{D}\sim\mathcal{P}^{n}}\mathbb{M}|Q_{1-\alpha}(V_{\mathcal{D}}^{o}/\|\nabla g(\hat{v})\|) - V_{\mathcal{D}}^{o}/\|\nabla g(\hat{v})\||^{\alpha}.\end{aligned} \tag{17}$$

According to Holder condition for quantile function, we obtain that $\mathbb{M}(\|\nabla g(\hat{v})\| \cdot Q_{1-\alpha}(V_{\mathcal{D}}^{o}/\|\nabla g(\hat{v})\|))$
$\leq \mathbb{M}V_{\mathcal{D}}^{o} + L\mathbb{M}|Q_{1-\alpha}(V_{\mathcal{D}}^{o}/\|\nabla g(\hat{v})\|) - V_{\mathcal{D}}^{o}/\|\nabla g(\hat{v})\||^{\alpha}$, therefore

$$\mathbb{E}_{\mathcal{D}}\mathbb{M}(\|\nabla g(\hat{v})\| \cdot Q_{1-\alpha}(V_{\mathcal{D}}^{o}/\|\nabla g(\hat{v})\|)) < \mathbb{E}_{\mathcal{D}}Q_{1-\alpha}(V_{\mathcal{D}}^{o}) - 2\max\{1, L\}[c/\sqrt{n}]^{\min\{1, \alpha\}}. \tag{18}$$

As the *Quantile Stability* assumption, we have that $\mathbb{E}_{\mathcal{D}\sim\mathcal{P}^{n}}|Q_{1-\alpha}(V_{\mathcal{D}}^{o}/\|\nabla g(\hat{v})\|) - Q_{1-\alpha}(V_{\mathcal{D}_{\mathrm{cal}}}^{o}/\|\nabla g(\hat{v}_{\mathrm{cal}})\|)|$
$\leq \frac{c}{\sqrt{n}}$ and $\mathbb{E}_{\mathcal{D}\sim\mathcal{P}^{n}}|Q_{1-\alpha}(V_{\mathcal{D}}^{o}) - Q_{1-\alpha}(V_{\mathcal{D}_{\mathrm{cal}}}^{o})| \leq \frac{c}{\sqrt{n}}$. Therefore,

$$\begin{aligned}&2\mathbb{E}(\|\nabla g(\hat{v})\| \cdot Q_{1-\alpha}(V_{\mathcal{D}_{\mathrm{cal}}}^{o}/\|\nabla g(\hat{v}_{\mathrm{cal}})\|) \\ &< 2Q_{1-\alpha}(V_{\mathcal{D}}^{o}) - 2\max\{1, L\}[c/\sqrt{n}]^{\min\{1, \alpha\}}, \\ &< 2Q_{1-\alpha}(V_{\mathcal{D}}^{o}).\end{aligned} \tag{19}$$

$\square$

# B Experimental Details

Section B.1 introduces the omitted experimental details. Section B.2 certifies the square conditions. Section B.3 discusses discusses the robustness of FFCP coverage with respect to the splitting point and across each network layer. Section B.4 demonstrates that FFCP performs similarly to Split CP in untrained neural networks, confirming that FFCP's efficiency is due to the semantic information trained in the feature space. Section B.5 proposes FFCQR after applying the gradient-level techniques of FFCP to CQR. Section B.6 proposes FFLCP after applying the gradient-level techniques of FFCP to LCP. Section B.7 proposes FFRAPS after applying the gradient-level techniques of FFCP to RAPS. Finally, Section B.8 provides additional experimental results.

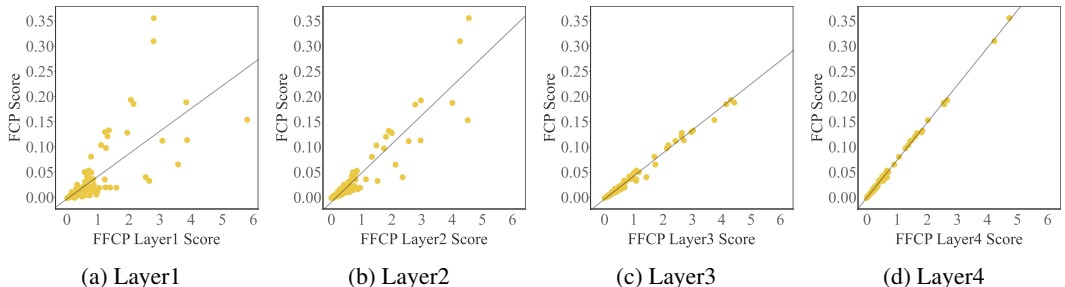

| (a) Layer1 | (b) Layer2 | (c) Layer3 | (d) Layer4 |

Figure 3: Scatter plot of FCP Score and FFCP Score at different layers. The relationship between the FCP Score and the FFCP Score is positively correlated, which indicates that the FFCP Score effectively approximates the FCP Score.

## B.1 Experimental Details

All the tests are performed on a desktop with an Intel Core i9-12900H CPU, NVIDIA GeForce RTX 4090 GPU, and 32 GB memory.

We conduct several more experiments in to establish the close relationship between FFCP and FCP, to demonstrate the benefits of FFCP from the good representation of gradient, and to provide empirical validations for the theoretical insights.

**Model Architecture.** For the one-dimensional we employ a four-layer neural network, with each layer consisting of 64 dimensions. For the semantic segmentation experiment, we utilize a network architecture combining ResNet50 with two additional convolutional layers. We use ResNet50 as the base feature extractor $h$, and the two subsequent convolution layers form the prediction head $g$.

**Correlation Between FCP and FFCP Scores Across Layers.** We compare the relationship between the scores of FCP and FFCP through experiments. Figure 3 indicates a positive correlation between the non-conformity scores of the two algorithms, suggesting that FFCP shares similarities with FCP in score function. This suggests that FFCP, while computationally efficient, provides a close approximation to FCP. The observed discrepancies may be attributed to the complex, layer-dependent non-linear transformations introduced by the decoder, under which the accuracy of the Taylor-based linear approximation tends to decline as the degree of non-linearity increases.

**Decoder Runtime Comparison.** We conducted additional experiments to evaluate how the decoder's layer depth influences computational efficiency. The running times presented below consider the decoder at various depths, explicitly showing the proportional relationship between decoder depth and computational cost.

In our primary comparison (FFCP vs. FCP), we maintained consistency by selecting the layer index as 2 for the FFCP, aligning it with the depth used by FCP. The detailed experimental results are summarized in Table 4:

Table 4: Running time comparison (mean $\pm$ std) for different decoder depths in STAR dataset.

| LAYER DEPTH | SPLIT CP | FCP | FFCP | FASTER |
|---|---|---|---|---|
| 2 LAYER | $0.0044\pm0.0009$ | $3.1302\pm0.4795$ | $0.0247\pm0.0053$ | 127X |
| 4 LAYER | $0.0047\pm0.0001$ | $5.8031\pm0.2011$ | $0.0239\pm0.0010$ | 243X |
| 6 LAYER | $0.0058\pm0.0003$ | $9.4411\pm0.5270$ | $0.0277\pm0.0011$ | 341X |
| 8 LAYER | $0.0064\pm0.0002$ | $12.2794\pm0.1668$ | $0.0294\pm0.0012$ | 418X |
| 10 LAYER | $0.0072\pm0.0001$ | $15.8716\pm0.3363$ | $0.0335\pm0.0015$ | 474X |
| 12 LAYER | $0.0077\pm0.0003$ | $19.4352\pm0.8947$ | $0.0373\pm0.0021$ | 521X |

These results demonstrate that while the computational cost of FCP grows significantly with the increasing depth of the decoder network, FFCP remains computationally efficient, only exhibiting marginal increases in runtime. Thus, FFCP provides substantial efficiency benefits, especially when employing deep network architectures.

**Robustness for FFCP.** The empirical performance of FFCP demonstrates its robustness, as seen in the ablation studies on splitting points. We demonstrate that coverage remains robust across different splitting points in neural networks, as detailed in Table 5 in Appendix B.3. Furthermore, the results from different layers of the FFCP network are consistent, as presented in Table 3

## B.2 Verifying Square Conditions

We verify the square conditions in this section. The key component of the square conditions is *Expansion* condition, which states that performing the quantile step does not result in a significant loss of efficiency.

For computational simplicity, We take exponent $\alpha = 1$ and do not consider the Lipschitz factor $L$. We next provide experiment results in Figure 4 on comparing the distribution of the scores between Split CP with FFCP.

From the figure, we observe that the overall distribution of FFCP non-conformity scores is closer to the quantile. This numerically validates that $\mathbf{M} \left| Q_{1-\alpha}(V_{\mathcal{D}}^o/|\nabla g(\hat{v})|) - V_{\mathcal{D}}^o/|\nabla g(\hat{v})| \right|$ is less than $\mathbf{M} \left[ Q_{1-\alpha}(V_{\mathcal{D}}^o) - V_{\mathcal{D}}^o \right]$.

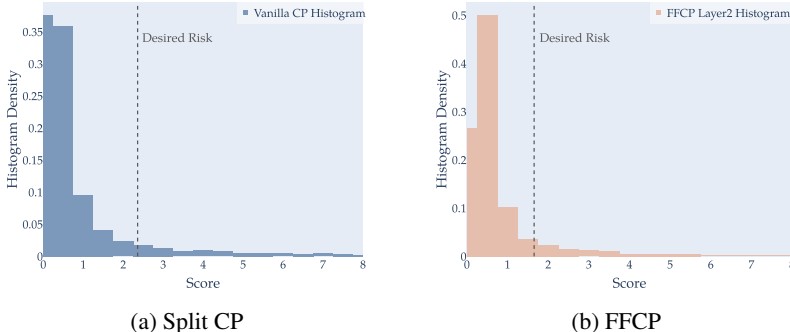

| (a) Split CP | (b) FFCP |
|---|---|

Figure 4: Empirical validation of Theorem A.1. We plot the score distributions and their corresponding quantiles ($\alpha = 0.1$) of Split CP (left) and FFCP (right). Compared to Split CP, the non-conformity scores of FFCP are closer to their quantiles, leading to a shorter band. Compared to Split, FFCP exhibits a more stable distribution with higher quantiles, leading to better performance for FFCP. FFCP selects layer 2 for display.

## B.3 Robustness of FFCP

To verify that the coverage by FFCP maintains its robustness despite changes in the splitting point, we performed a network split. The experimental results, detailed in Table 5, demonstrate that FFCP is indeed robust.

Table 5: Ablation study of the number of layers in $h$ and $g$ in unidimensional tasks. For the sake of avoiding redundancy, we set $\alpha = 0.05$.

| DATASET | | FACEBOOK1 | | MEPS19 | | BLOG | |
|---|---|---|---|---|---|---|---|
| METHOD | NUMBER($g \circ h$) | COVERAGE | LENGTH | COVERAGE | LENGTH | COVERAGE | LENGTH |
| SPLIT CP | / | $95.24 \pm 0.16$ | $4.60 \pm 0.50$ | $95.35 \pm 0.23$ | $7.34 \pm 1.01$ | $95.08 \pm 0.11$ | $7.88 \pm 0.97$ |
| FFCP | $h:0 \quad g:4$ | $94.88 \pm 0.19$ | $5.35 \pm 0.42$ | $95.18 \pm 0.40$ | $5.36 \pm 0.52$ | $94.97 \pm 0.31$ | $8.28 \pm 0.59$ |
| | $h:1 \quad g:3$ | $94.84 \pm 0.14$ | $4.21 \pm 0.46$ | $95.13 \pm 0.38$ | $5.15 \pm 0.49$ | $94.99 \pm 0.18$ | $6.37 \pm 1.22$ |
| | $h:2 \quad g:2$ | $95.14 \pm 0.13$ | $2.48 \pm 0.09$ | $95.19 \pm 0.36$ | $5.57 \pm 0.87$ | $95.08 \pm 0.12$ | $5.36 \pm 0.93$ |
| | $h:3 \quad g:1$ | $95.16 \pm 0.18$ | $2.59 \pm 0.72$ | $95.34 \pm 0.22$ | $6.95 \pm 1.13$ | $95.05 \pm 0.11$ | $6.46 \pm 1.52$ |
| | $h:4 \quad g:0$ | $95.24 \pm 0.16$ | $4.60 \pm 0.50$ | $95.35 \pm 0.23$ | $7.34 \pm 1.01$ | $95.08 \pm 0.11$ | $7.88 \pm 0.97$ |

Table 6: Untrained model comparison between Split CP and FFCP. When the model has not been sufficiently trained, FFCP performs similarly to Split CP. This means that the model's performance determines the quality of the feature information in the gradient. When the model performs poorly, the gradient information obtained by FFCP is inaccurate. On the other hand, this also suggests that FFCP effectively utilizes the feature information in the gradient when the model is well-trained.

| METHOD | SPLIT CP | | FFCP | |
|---|---|---|---|---|
| DATASET | COVERAGE | LENGTH | COVERAGE | LENGTH |
| SYNTHETIC | $90.23\pm0.45$ | $\mathbf{2.34}\pm0.01$ | $90.22\pm0.96$ | $2.41\pm0.01$ |
| COM | $90.33\pm1.81$ | $4.86\pm0.13$ | $90.43\pm1.99$ | $\mathbf{4.73}\pm0.08$ |
| FB1 | $90.18\pm0.19$ | $3.57\pm0.09$ | $90.10\pm0.13$ | $\mathbf{3.57}\pm0.08$ |
| FB2 | $90.16\pm0.11$ | $3.66\pm0.11$ | $90.12\pm0.14$ | $\mathbf{3.66}\pm0.06$ |
| MEPS19 | $90.80\pm0.43$ | $\mathbf{4.33}\pm0.07$ | $90.85\pm0.58$ | $4.38\pm0.07$ |
| MEPS20 | $90.15\pm0.55$ | $\mathbf{4.41}\pm0.23$ | $90.27\pm0.63$ | $4.46\pm0.25$ |
| MEPS21 | $89.80\pm0.45$ | $\mathbf{4.41}\pm0.17$ | $89.89\pm0.56$ | $\mathbf{4.41}\pm0.15$ |
| STAR | $89.79\pm0.51$ | $\mathbf{1.88}\pm0.01$ | $89.98\pm0.56$ | $1.94\pm0.01$ |
| BIO | $90.16\pm0.20$ | $4.09\pm0.02$ | $90.07\pm0.14$ | $\mathbf{4.04}\pm0.02$ |
| BLOG | $90.11\pm0.30$ | $\mathbf{2.53}\pm0.12$ | $90.12\pm0.28$ | $2.55\pm0.14$ |
| BIKE | $89.55\pm0.82$ | $\mathbf{4.56}\pm0.09$ | $89.57\pm0.86$ | $4.60\pm0.10$ |

Table 7: Untrained model comparison between Split CP and FFCP by Layer on Each Dataset (lower is better).

| DATASET | LAYER1 | LAYER2 | LAYER3 | LAYER4 | LAYER5 (SPLIT) |
|---|---|---|---|---|---|
| SYNTHETIC | $2.87\pm0.03$ | $2.81\pm0.03$ | $2.89\pm0.01$ | $2.41\pm\mathbf{0.01}$ | $\mathbf{2.34}\pm0.01$ |
| COM | $5.41\pm0.14$ | $5.30\pm0.18$ | $5.03\pm0.09$ | $\mathbf{4.73}\pm\mathbf{0.08}$ | $4.86\pm0.13$ |
| FB1 | $3.90\pm0.01$ | $3.77\pm0.10$ | $3.62\pm0.08$ | $3.56\pm0.08$ | $\mathbf{3.56}\pm\mathbf{0.09}$ |
| FB2 | $4.02\pm0.09$ | $3.90\pm0.8$ | $3.74\pm0.08$ | $3.66\pm0.06$ | $\mathbf{3.66}\pm\mathbf{0.11}$ |
| MEPS19 | $4.38\pm0.06$ | $4.41\pm0.07$ | $4.42\pm0.07$ | $4.38\pm0.07$ | $\mathbf{4.33}\pm\mathbf{0.07}$ |
| MEPS20 | $4.43\pm0.21$ | $4.44\pm0.23$ | $4.48\pm0.25$ | $4.46\pm0.25$ | $4.41\pm0.23$ |
| MEPS21 | $\mathbf{4.41}\pm\mathbf{0.18}$ | $4.46\pm0.18$ | $4.48\pm0.18$ | $4.41\pm0.15$ | $4.41\pm0.17$ |
| STAR | $2.49\pm0.06$ | $2.40\pm0.05$ | $2.15\pm0.03$ | $1.94\pm0.01$ | $\mathbf{1.88}\pm\mathbf{0.01}$ |
| BIO | $4.27\pm0.02$ | $4.14\pm0.02$ | $4.07\pm0.02$ | $\mathbf{4.04}\pm\mathbf{0.02}$ | $4.08\pm0.02$ |
| BLOG | $2.56\pm0.15$ | $2.54\pm0.15$ | $2.57\pm0.13$ | $2.55\pm0.14$ | $\mathbf{2.53}\pm\mathbf{0.12}$ |
| BIKE | $4.69\pm0.09$ | $4.67\pm0.07$ | $4.57\pm0.09$ | $4.60\pm0.10$ | $\mathbf{4.56}\pm\mathbf{0.09}$ |

## B.4 FFCP works due to semantic information in feature space

One of our primary advantages is that FFCP leverages the semantic information of gradient in feature space. This is due to the fact that gradient-level techniques in feature space improve efficiency via the robust feature embedding abilities of well-trained neural networks.

On the other hand, when the base model is untrained and initialized randomly, lacking meaningful semantic representation in gradient, the band length produced by FFCP is comparable to Split CP. For results, see Table 6.

**FFCP on untrained network.** We propose that FFCP returns shorter band lengths through its deployment of deep representations from the gradients. To test this view, we contrast FFCP's performance using an untrained neural network against a baseline model. Using an incompletely trained neural network, FFCP's performance deteriorates and becomes comparable to that of Split CP. This is due to the partially incorrect semantic information in the gradient, which *misleads* FFCP. We defer the results to Table 6 and have updated the results in Table 7 by selecting the best-performing layer, which confirms that FFCP underperforms when the model is not sufficiently trained.

## B.5 FFCQR

This section highlights the adaptability of FFCP's gradient-level techniques, showing their suitability for a wide range of existing conformal prediction algorithms. We choose Conformalized Quantile Regression (CQR, Romano et al. [2019b]) to propose Fast Feature Conformalized Quantile Regression (FFCQR). The fundamental concept is similar to FFCP Algorithm 2, where calibration steps are performed within the gradient information. FFCQR algorithm is proposed in Algorithm 3.

---

**Algorithm 3** Fast Feature Conformalized Quantile Regression (FFCQR)

---

**Input:** Confidence level $\alpha$, dataset $\mathcal{D} = \{(X_i, Y_i)\}_{i \in \mathcal{I}}$, test point $X'$;

1: Randomly split the dataset $\mathcal{D}$ into a training fold $\mathcal{D}_{\text{tra}} \triangleq (X_i, Y_i)_{i \in \mathcal{I}_{\text{tra}}}$ together with a calibration fold $\mathcal{D}_{\text{cal}} \triangleq (X_i, Y_i)_{i \in \mathcal{I}_{\text{cal}}}$;

2: Train a base machine learning model $f^{\text{lo}} = g^{\text{lo}} \circ h(\cdot)$ and $f^{\text{hi}} = g^{\text{hi}} \circ h(\cdot)$ using $\mathcal{D}_{\text{tra}}$ to estimate the quantile of response $Y_i$, which returns $[f^{\text{lo}}(X_i), f^{\text{hi}}(X_i)]$;

3: For each $i \in \mathcal{I}_{\text{cal}}$, calculate the non-conformity score $\tilde{R}_i^{\text{lo}} = (f^{\text{lo}}(X_i) - Y_i)/\|\nabla g^{\text{lo}}(\hat{v}_i)\|$ and $\tilde{R}_i^{\text{hi}} = (Y_i - f^{\text{hi}}(X_i))/\|\nabla g^{\text{hi}}(\hat{v}_i)\|$, where $\nabla g(\cdot)$ denote the gradient of $g(\cdot)$ on the feature $\hat{v}_i \triangleq h(X_i)$, namely $\nabla g^{\text{lo}}(\hat{v}_i) = \frac{dg^{\text{lo}} \circ h(X_i)}{dh(X_i)}$ and $\nabla g^{\text{hi}}(\hat{v}_i) = \frac{dg^{\text{hi}} \circ h(X_i)}{dh(X_i)}$

4: Calculate the $(1 - \alpha/2)$-th quantile $Q_{1-\alpha/2}$ of the distribution $\frac{1}{|\mathcal{I}_{\text{cal}}|+1} \sum_{i \in \mathcal{I}_{\text{cal}}} \delta_{\tilde{R}_i} + \delta_\infty$, where $\tilde{R}_i = \max\left\{\tilde{R}_i^{\text{lo}}, \tilde{R}_i^{\text{hi}}\right\}$

**Output:** $\mathcal{C}_{1-\alpha/2}^{\text{ffcqr}}(X') = \left[f^{\text{lo}}(X') - \|\nabla g^{\text{lo}}(\hat{v}')\| \cdot Q_{1-\alpha/2}, f^{\text{hi}}(X') + \|\nabla g^{\text{hi}}(\hat{v}')\| \cdot Q_{1-\alpha/2}\right]$, where $\hat{v}' = h(X')$.

---

We summarize run time in Table 8 and the experiments result in Table 9 (meps19), Table 10 (com), and Table 11 (bike). FFCQR reduces runtime compared to FCQR, while achieving better efficiency compared to CQR.

Furthermore, we have observed that as the values of $[\alpha, 1 - \alpha]$ used by the neural networks in all CQR methods (CQR, FCQR and FFCQR) become increasingly closer in the training process (The level difference between [0.1, 0.9] is 0.8, while the level difference between [0.49, 0.51] is 0.02, with the difference gradually decreasing), the band length returned by FFCQR gradually narrows. This implies that our method holds an advantage on returned band length when the narrower neural network confidence level.

Table 8: Time Comparison among CQR, FCQR and FFCQR. For quantile regression tasks, FFCQR also demonstrates more efficient performance. The last column represents the speed improvement factor of FFCQR compared to FCQR. The time unit is in seconds.

| DATASET | CQR | FCQR | FFCQR | FASTER |
|---------|-----|------|-------|--------|
| SYNTHETIC | 0.0125±0.0062 | 0.3237±0.0152 | 0.0742±0.0091 | 4X |
| COM | 0.0045±0.0015 | 0.2730±0.1088 | 0.0210±0.0011 | 13X |
| FB1 | 0.0446±0.0157 | 1.7276±0.1389 | 0.2532±0.0166 | 7X |
| FB2 | 0.0812±0.0187 | 3.9967±0.7330 | 0.0617±0.0123 | 65X |
| MEPS19 | 0.0187±0.0018 | 0.7671±0.0438 | 0.1189±0.0048 | 6X |
| MEPS20 | 0.0438±0.0079 | 1.1876±0.2206 | 0.1505±0.0138 | 8X |
| MEPS21 | 0.0187±0.0027 | 0.8004±0.0657 | 0.1120±0.0053 | 7X |
| STAR | 0.0047±0.0009 | 0.2352±0.0419 | 0.0214±0.0005 | 11X |
| BIO | 0.0774±0.0541 | 6.9365±4.5494 | 0.6473±0.4879 | 11X |
| BLOG | 0.1121±0.0153 | 1.9591±0.1346 | 0.3941±0.0618 | 5X |
| BIKE | 0.0138±0.0045 | 1.8528±2.3969 | 0.2382±0.3261 | 6X |

Table 9: Coverage and Band Length at Different Net Confidence Levels Used By the Neural Networks in CQR methods with *Meps19* dataset. FFCQR yields shorter band lengths compared to CQR.

| CONFIDENCE LEVELS | | [0.1, 0.9] | | [0.2, 0.8] | | [0.3, 0.7] | | [0.4, 0.6] | | [0.49, 0.51] | |
|---|---|---|---|---|---|---|---|---|---|---|---|---|
| METRICS | | COVERAGE | LENGTH | COVERAGE | LENGTH | COVERAGE | LENGTH | COVERAGE | LENGTH | COVERAGE | LENGTH |
| SPLIT CQR | | $90.28 \pm 0.47$ | $2.43 \pm 0.11$ | $90.19 \pm 0.46$ | $2.58 \pm 0.45$ | $90.63 \pm 0.32$ | $2.93 \pm 0.51$ | $90.48 \pm 0.42$ | $3.47 \pm 0.16$ | $90.44 \pm 0.36$ | $3.48 \pm 0.08$ |
| FCQR | | $91.32 \pm 0.37$ | $1.50 \pm 0.37$ | $90.26 \pm 0.33$ | $2.61 \pm 2.01$ | $90.45 \pm 0.54$ | $2.30 \pm 2.38$ | $90.58 \pm 0.33$ | $6.11 \pm 0.89$ | $90.47 \pm 0.45$ | $4.59 \pm 1.73$ |
| | LAYER0 | $90.29 \pm 0.60$ | $2.61 \pm 0.13$ | $90.22 \pm 0.26$ | $2.58 \pm 0.54$ | $90.31 \pm 0.43$ | $3.03 \pm 0.90$ | $89.95 \pm 0.29$ | $5.30 \pm 0.57$ | $89.84 \pm 0.12$ | $5.96 \pm 1.26$ |
| | LAYER1 | $90.29 \pm 0.57$ | $2.56 \pm 0.13$ | $90.10 \pm 0.33$ | $2.49 \pm 0.52$ | $90.34 \pm 0.46$ | $2.92 \pm 0.85$ | $90.02 \pm 0.33$ | $5.07 \pm 0.50$ | $89.88 \pm 0.24$ | $5.71 \pm 1.22$ |
| FFCQR | LAYER2 | $90.14 \pm 0.60$ | $2.34 \pm 0.12$ | $90.24 \pm 0.65$ | $2.22 \pm 0.32$ | $90.34 \pm 0.43$ | $2.60 \pm 0.68$ | $89.96 \pm 0.40$ | $4.10 \pm 0.22$ | $89.97 \pm 0.33$ | $4.76 \pm 1.17$ |
| | LAYER3 | $90.21 \pm 0.45$ | $2.18 \pm 0.12$ | $90.14 \pm 0.42$ | $2.10 \pm 0.19$ | $90.35 \pm 0.36$ | $2.34 \pm 0.19$ | $90.28 \pm 0.33$ | $2.76 \pm 0.15$ | $89.86 \pm 0.49$ | $3.16 \pm 0.62$ |
| | LAYER4 | $90.28 \pm 0.47$ | $2.43 \pm 0.11$ | $90.19 \pm 0.46$ | $2.58 \pm 0.45$ | $90.63 \pm 0.32$ | $2.93 \pm 0.51$ | $90.48 \pm 0.42$ | $3.47 \pm 0.16$ | $90.44 \pm 0.36$ | $3.48 \pm 0.08$ |

Table 10: Coverage and Band Length at Different Net Confidence Levels Used By the Neural Networks in CQR methods with *com* dataset. FFCQR yields shorter band lengths compared to CQR.

| CONFIDENCE LEVELS | | [0.1, 0.9] | | [0.2, 0.8] | | [0.3, 0.7] | | [0.4, 0.6] | | [0.49, 0.51] | |
|---|---|---|---|---|---|---|---|---|---|---|---|---|
| METRICS | | COVERAGE | LENGTH | COVERAGE | LENGTH | COVERAGE | LENGTH | COVERAGE | LENGTH | COVERAGE | LENGTH |
| SPLIT CQR | | $89.87 \pm 1.68$ | $1.57 \pm 0.12$ | $90.13 \pm 0.89$ | $1.71 \pm 0.18$ | $89.87 \pm 1.06$ | $1.74 \pm 0.16$ | $89.27 \pm 0.94$ | $2.07 \pm 0.55$ | $89.57 \pm 0.49$ | $1.99 \pm 0.12$ |
| FCQR | | $90.83 \pm 1.53$ | $1.19 \pm 0.19$ | $90.43 \pm 1.32$ | $0.49 \pm 0.38$ | $90.23 \pm 1.13$ | $0.37 \pm 0.06$ | $90.18 \pm 1.77$ | $0.20 \pm 0.05$ | $89.47 \pm 0.85$ | $0.23 \pm 0.07$ |
| | LAYER0 | $88.92 \pm 2.78$ | $1.62 \pm 0.12$ | $89.62 \pm 1.75$ | $1.67 \pm 0.07$ | $91.53 \pm 0.97$ | $1.62 \pm 0.12$ | $89.77 \pm 1.64$ | $1.80 \pm 0.27$ | $89.82 \pm 1.07$ | $1.76 \pm 0.11$ |
| | LAYER1 | $88.67 \pm 2.40$ | $1.59 \pm 0.12$ | $89.57 \pm 1.03$ | $1.64 \pm 0.08$ | $90.58 \pm 1.21$ | $1.57 \pm 0.13$ | $89.82 \pm 1.75$ | $1.78 \pm 0.33$ | $89.12 \pm 1.40$ | $1.74 \pm 0.09$ |
| FFCQR | LAYER2 | $89.77 \pm 2.14$ | $1.58 \pm 0.12$ | $89.92 \pm 1.98$ | $1.63 \pm 0.12$ | $90.53 \pm 0.43$ | $1.64 \pm 0.14$ | $89.67 \pm 1.28$ | $1.89 \pm 0.38$ | $88.77 \pm 0.75$ | $1.78 \pm 0.11$ |
| | LAYER3 | $90.08 \pm 2.28$ | $1.58 \pm 0.12$ | $89.92 \pm 1.22$ | $1.67 \pm 0.15$ | $90.33 \pm 0.86$ | $1.73 \pm 0.13$ | $89.62 \pm 0.83$ | $2.03 \pm 0.54$ | $89.27 \pm 0.66$ | $1.93 \pm 0.11$ |
| | LAYER4 | $89.87 \pm 1.68$ | $1.57 \pm 0.12$ | $90.13 \pm 0.89$ | $1.71 \pm 0.18$ | $89.87 \pm 1.06$ | $1.74 \pm 0.16$ | $89.27 \pm 0.94$ | $2.07 \pm 0.55$ | $89.57 \pm 0.49$ | $1.99 \pm 0.12$ |

## B.6 Group coverage

*Group coverage* is represented by the conditional probability $\mathbb{P}(Y \in \mathcal{C}(X)|X)$. The test dataset was categorized into three groups by splitting response $Y$ based on the lower and upper tertiles, and we have reported the minimum coverage for each group.

We present our results in two parts: (a) we present the group coverage provided by Split CP, FCP, FFCP, detailed in Table 14 and (b) the group coverage provided by LCP and FFLCP, as shown in Table 13.

Analyzing the experimental results, we believe that the group coverage achieved through gradient-level techniques in FFCP reflects an improvement over Split CP, albeit with moderate overall performance. We note that the group coverage of gradient-level conformal prediction is contingent upon its Split version. That is, when the Split version demonstrates satisfying group coverage, the gradient-level version tends to mirror this result. Thus, despite FFCP outperforming Split CP, the overall performance is still considered average.

LCP, developed specifically to enhance group coverage, inherently achieves higher coverage. Experimental results further reveal that FFLCP surpasses LCP, demonstrating the superiority of our gradient-level techniques.

---

**Algorithm 4** Fast Feature Localized Conformal Prediction (FFLCP)

**Input:** Confidence level $\alpha$, dataset $\mathcal{D} = \{(X_i, Y_i)\}_{i \in \mathcal{I}}$, tesing point $X'$, localizer $D(X, Y)$
1: Randomly split the dataset $\mathcal{D}$ into a training fold $\mathcal{D}_{\text{tra}} \triangleq \{(X_i, Y_i)\}_{i \in \mathcal{I}_{\text{tra}}}$ and a calibration fold $\mathcal{D}_{\text{cal}} \triangleq \{(X_i, Y_i)\}_{i \in \mathcal{I}_{\text{cal}}}$ ;
2: Train a base neural network with training fold $f(\cdot) = g \circ h(\cdot)$ with training fold $\mathcal{D}_{\text{tra}}$;
3: For each $i \in \mathcal{I}_{\text{cal}}$, calculate the non-conformity score $\tilde{R}_i = |Y_i - f(X_i)|/\|\nabla g(\hat{v}_i)\|$, where $\nabla g(\hat{v}_i)$ denotes the gradient of $g(\cdot)$ on the feature $\hat{v}_i \triangleq h(X_i)$, namely $\nabla g(\hat{v}_i) = \frac{\text{d}g \circ h(X_i)}{\text{d}h(X_i)}$;
4: Calculate the distance $D_i \triangleq D(X', X_i)$, $d_i^D := \frac{D_i}{\sum_{i \in \mathcal{I}_{\text{cal}}} D_i}$ and $(1 - \alpha)$-th quantile $Q_{1-\alpha}$ of the distribution $\sum_{i \in \mathcal{I}_{\text{cal}}} d_i^D \delta_{\tilde{R}_i} + \delta_\infty$;
**Output:** $\mathcal{C}_{1-\alpha}^{\text{fflcp}}(X') = [f(X') - \|\nabla g(\hat{v}')\|Q_{1-\alpha}, f(X') + \|\nabla g(\hat{v}')\|Q_{1-\alpha}]$, where $\hat{v}' = h(X')$.

Table 11: Coverage and Band Length at Different Net Confidence Levels Used By the Neural Networks in CQR methods with *bike* dataset. FFCQR yields shorter band lengths compared to CQR.

| CONFIDENCE LEVELS | | [0.1, 0.9] | | [0.2, 0.8] | | [0.3, 0.7] | | [0.4, 0.6] | | [0.49, 0.51] | |
|---|---|---|---|---|---|---|---|---|---|---|---|
| METRICS | | COVERAGE | LENGTH | COVERAGE | LENGTH | COVERAGE | LENGTH | COVERAGE | LENGTH | COVERAGE | LENGTH |
| SPLIT CQR | | $89.38_{\pm0.73}$ | $0.82_{\pm0.07}$ | $89.99_{\pm0.69}$ | $0.73_{\pm0.03}$ | $89.63_{\pm0.84}$ | $0.77_{\pm0.03}$ | $90.25_{\pm0.62}$ | $0.84_{\pm0.02}$ | $89.72_{\pm0.51}$ | $0.96_{\pm0.08}$ |
| FCQR | | $90.25_{\pm0.67}$ | $0.58_{\pm0.15}$ | $90.14_{\pm0.63}$ | $0.65_{\pm0.13}$ | $89.77_{\pm0.76}$ | $0.71_{\pm0.18}$ | $89.93_{\pm0.35}$ | $0.82_{\pm0.07}$ | $89.98_{\pm0.97}$ | $0.74_{\pm0.08}$ |
| FFCQR | LAYER0 | $89.91_{\pm0.38}$ | $0.91_{\pm0.07}$ | $89.84_{\pm0.44}$ | $0.97_{\pm0.02}$ | $89.42_{\pm0.83}$ | $1.20_{\pm0.04}$ | $89.75_{\pm0.54}$ | $1.64_{\pm0.13}$ | $89.61_{\pm0.59}$ | $1.79_{\pm0.08}$ |
| | LAYER1 | $89.57_{\pm0.33}$ | $0.90_{\pm0.07}$ | $89.83_{\pm0.25}$ | $0.90_{\pm0.05}$ | $89.44_{\pm0.42}$ | $1.04_{\pm0.07}$ | $89.72_{\pm0.43}$ | $1.25_{\pm0.05}$ | $89.62_{\pm0.73}$ | $1.31_{\pm0.06}$ |
| | LAYER2 | $89.73_{\pm0.29}$ | $0.87_{\pm0.07}$ | $89.70_{\pm0.27}$ | $0.79_{\pm0.03}$ | $89.63_{\pm0.69}$ | $0.83_{\pm0.03}$ | $89.14_{\pm0.42}$ | $0.92_{\pm0.04}$ | $89.44_{\pm0.41}$ | $0.98_{\pm0.04}$ |
| | LAYER3 | $89.49_{\pm0.34}$ | $0.84_{\pm0.06}$ | $89.62_{\pm0.48}$ | $0.69_{\pm0.02}$ | $89.58_{\pm0.74}$ | $0.69_{\pm0.02}$ | $89.86_{\pm0.37}$ | $0.70_{\pm0.01}$ | $89.57_{\pm0.88}$ | $0.78_{\pm0.07}$ |
| | LAYER4 | $89.38_{\pm0.73}$ | $0.82_{\pm0.07}$ | $89.99_{\pm0.69}$ | $0.73_{\pm0.03}$ | $89.63_{\pm0.84}$ | $0.77_{\pm0.03}$ | $90.25_{\pm0.62}$ | $0.84_{\pm0.02}$ | $89.72_{\pm0.51}$ | $0.96_{\pm0.08}$ |

Table 12: Coverage and Band Length at Different Net Confidence Levels Used By the Neural Networks in CQR methods with *bio* dataset. FFCQR yields shorter band lengths compared to CQR.

| CONFIDENCE LEVELS | | [0.1, 0.9] | | [0.2, 0.8] | | [0.3, 0.7] | | [0.4, 0.6] | | [0.49, 0.51] | |
|---|---|---|---|---|---|---|---|---|---|---|---|
| METRICS | | COVERAGE | LENGTH | COVERAGE | LENGTH | COVERAGE | LENGTH | COVERAGE | LENGTH | COVERAGE | LENGTH |
| SPLIT CQR | | $89.89_{\pm0.41}$ | $1.42_{\pm0.02}$ | $89.84_{\pm0.27}$ | $1.45_{\pm0.02}$ | $89.87_{\pm0.27}$ | $1.61_{\pm0.02}$ | $90.07_{\pm0.31}$ | $1.86_{\pm0.03}$ | $90.16_{\pm0.40}$ | $2.00_{\pm0.03}$ |
| FCQR | | $90.18_{\pm0.35}$ | $0.95_{\pm0.50}$ | $90.45_{\pm0.45}$ | $2.09_{\pm0.41}$ | $90.16_{\pm0.48}$ | $1.84_{\pm0.43}$ | $90.25_{\pm0.46}$ | $2.37_{\pm0.76}$ | $90.21_{\pm0.46}$ | $2.02_{\pm0.34}$ |
| FFCQR | LAYER0 | $89.74_{\pm0.32}$ | $1.47_{\pm0.01}$ | $89.98_{\pm0.22}$ | $1.56_{\pm0.04}$ | $89.89_{\pm0.25}$ | $1.73_{\pm0.04}$ | $89.87_{\pm0.24}$ | $2.22_{\pm0.15}$ | $89.64_{\pm0.20}$ | $2.55_{\pm0.06}$ |
| | LAYER1 | $89.77_{\pm0.33}$ | $1.45_{\pm0.01}$ | $89.99_{\pm0.21}$ | $1.48_{\pm0.03}$ | $89.92_{\pm0.37}$ | $1.59_{\pm0.03}$ | $89.92_{\pm0.21}$ | $1.99_{\pm0.12}$ | $89.69_{\pm0.28}$ | $2.21_{\pm0.04}$ |
| | LAYER2 | $89.77_{\pm0.40}$ | $1.43_{\pm0.02}$ | $90.01_{\pm0.23}$ | $1.41_{\pm0.01}$ | $90.02_{\pm0.32}$ | $1.49_{\pm0.03}$ | $90.01_{\pm0.49}$ | $1.76_{\pm0.11}$ | $89.79_{\pm0.35}$ | $1.94_{\pm0.07}$ |
| | LAYER3 | $89.75_{\pm0.41}$ | $1.41_{\pm0.02}$ | $89.98_{\pm0.34}$ | $1.38_{\pm0.02}$ | $89.93_{\pm0.41}$ | $1.47_{\pm0.01}$ | $90.07_{\pm0.12}$ | $1.68_{\pm0.04}$ | $89.97_{\pm0.34}$ | $1.78_{\pm0.02}$ |
| | LAYER4 | $89.89_{\pm0.41}$ | $1.42_{\pm0.02}$ | $89.84_{\pm0.27}$ | $1.45_{\pm0.02}$ | $89.87_{\pm0.27}$ | $1.61_{\pm0.02}$ | $90.07_{\pm0.31}$ | $1.86_{\pm0.03}$ | $90.16_{\pm0.40}$ | $2.00_{\pm0.03}$ |

## B.7 FFRAPS

In this section, we show how to deploy gradient-level techniques in FFCP in classification problems. The basic ideas follow Algorithm 5.

Comparing to the experimental part of RAPS, our core adjustments are as follows:

(a) During the calibration process, for the model's output of sorted scores $s$, we divide each element by the magnitude of its corresponding gradient: $s + \delta \cdot \|\nabla g(v)\|$. Here, $\delta$ is an adjustable hyper-parameter that can be tuned to optimize the performance of the model based on the specific characteristics of the data and the problem at hand.

(b) In the stage of calculating the returned set, we multiply the generalized inverse quantile $\tau$ by the magnitude of the gradient of the corresponding test data: $s' + \delta \cdot \|\nabla g(v')\|$

We summarize the experiment results in Table 15, where we adhere to the statistical methodologies of RAPS as described in Angelopoulos et al. [2020].

---

**Algorithm 5** Fast Feature Regularized Adaptive Prediction Sets (FFRAPS)

**Input:** Confidence level $\alpha$, dataset $\mathcal{D} = \{(X_i, Y_i)\}_{i \in \mathcal{I}}$, tesing point $X'$, and ground-truth label $y \in \{0, 1, ..., K\}^n$ for $X \in \mathcal{D}$ and $X'$; regularization hyperparameters $k_{reg}$, $\delta$ and $\lambda$;

1: Randomly split the dataset $\mathcal{D}$ into a training fold $\mathcal{D}_{\text{tra}} \triangleq \{(X_i, Y_i)\}_{i \in \mathcal{I}_{\text{tra}}}$ and a calibration fold $\mathcal{D}_{\text{cal}} \triangleq \{(X_i, Y_i)\}_{i \in \mathcal{I}_{\text{cal}}}$ ;

2: Train a base neural network with training fold $f(\cdot) = g \circ h(\cdot)$ with training fold $\mathcal{D}_{\text{tra}}$;

3: For each $i \in \mathcal{I}_{\text{cal}}$, $L_i \leftarrow j$ such that $I_{i,j} = y_i$, where $I$ represents the associated permutation of index. Calculate generalized inverse quantile conformity score $E_i = \Sigma_{j=1}^{L_i} s_{i,j} + \|\nabla g(\hat{v}_i)\| \cdot \delta + \lambda(L_i - k_{reg})^+$ , where $\nabla g(\hat{v}_i)$ denotes the gradient of $g(\cdot)$ on the feature $\hat{v}_i \triangleq h(X_i)$, namely $\nabla g(\hat{v}_i) = \frac{\mathrm{d}g \circ h(X_i)}{\mathrm{d}h(X_i)}$, where $s \triangleq \mathrm{sort} f(X)$ represents the sorted scores. Calculate $\hat{\tau}_{ccal} \leftarrow \lceil(1 - \alpha)(1 + n)\rceil$ largest value in $\{E_i\}_{i=1}^n$

4: Calculate $L \leftarrow | \{j \in \mathcal{Y} : \Sigma_{i=1}^j s_i' + \|\nabla g(\hat{v}_i')\| \cdot \delta + \lambda(j - k_{reg})^+ \leq \hat{\tau}_{ccal}*\} | + 1$, where $\hat{v}' = h(X')$ and $s' = \mathrm{sort} f(X')$;

**Output:** $\mathcal{C}_{1-\alpha}^{\mathrm{FFRAPS}}(X') = \{I_1, ...I_L\}$

---

Table 13: Comparison of LCP and FFLCP in group coverage. We divide the datasets into three groups based on the size of $Y$, and calculate the coverage for each group, returning the maximum coverage. FFLCP shows the results for the 5-layer neural network.

| METHOD | LCP | FFLCP | | | | |
|---|---|---|---|---|---|---|
| DATASET | COVERAGE | LAYER0 | LAYER1 | LAYER2 | LAYER3 | LAYER4 |
| SYNTHETIC | 87.02±1.00 | 86.93±0.78 | 86.57±0.88 | 85.43±1.24 | **87.11**±1.76 | 87.02±1.00 |
| COM | 80.33±3.24 | **81.84**±2.52 | 79.56±3.34 | 77.42±4.12 | 79.41±2.88 | 80.33±3.24 |
| FB1 | 52.51±1.76 | **78.61**±0.91 | 76.39±1.21 | 67.16±1.61 | 57.82±1.88 | 52.51±1.76 |
| FB2 | 54.33±1.75 | **75.86**±0.83 | 75.44±0.91 | 70.45±0.99 | 60.18±1.73 | 54.33±1.75 |
| MEPS19 | 67.35±1.21 | **68.19**±2.31 | 66.44±2.01 | 64.94±1.53 | 67.14±1.24 | 67.35±1.21 |
| MEPS20 | 65.49±1.64 | **69.30**±1.09 | 68.80±1.55 | 65.14±1.44 | 65.47±1.99 | 65.49±1.64 |
| MEPS21 | 66.38±0.95 | 67.82±1.10 | **67.96**±1.21 | 66.21±1.33 | 65.54±1.02 | 66.38±0.95 |
| STAR | 77.20±3.97 | **79.69**±2.88 | 79.28±1.72 | 77.47±5.21 | 77.33±4.12 | 77.20±3.97 |
| BIO | 81.10±0.61 | 86.33±0.51 | 86.06±0.50 | **86.78**±0.57 | 83.26±0.71 | 81.10±0.61 |
| BLOG | 48.99±1.01 | **61.01**±0.82 | 55.10±1.12 | 46.01±0.65 | 46.88±0.81 | 48.99±1.01 |
| BIKE | 77.61±1.52 | 81.02±1.73 | 82.42±2.08 | 82.97±1.29 | **84.41**±1.71 | 77.61±1.52 |

Table 14: Comparison of Split CP, FCP and FFCP in group coverage.

| METHOD | SPLIT CP | FCP | FFCP | | | | |
|---|---|---|---|---|---|---|---|
| DATASET | COVERAGE | COVERAGE | LAYER0 | LAYER1 | LAYER2 | LAYER3 | LAYER4 |
| SYNTHETIC | 87.08±1.03 | 87.92±1.08 | 86.96±0.81 | 86.63±0.79 | 85.64±1.13 | **88.46**±1.44 | 87.08±1.03 |
| COM | 79.41±3.12 | 79.57±2.96 | **82.00**±3.18 | 79.41±3.62 | 78.64±4.35 | 78.65±3.62 | 79.41±3.12 |
| FB1 | 56.69±1.35 | 57.34±1.12 | **79.20**±0.95 | 76.75±1.42 | 68.09±1.76 | 59.33±1.91 | 56.69±1.35 |
| FB2 | 57.98±1.28 | 58.72±0.87 | **76.27**±0.92 | 75.64±0.91 | 70.86±0.89 | 62.43±1.15 | 57.98±1.28 |
| MEPS19 | 73.78±1.08 | **73.82**±0.91 | 70.90±2.29 | 70.51±2.28 | 72.09±1.25 | 73.53±1.00 | 73.78±1.08 |
| MEPS20 | 72.21±1.47 | **72.33**±1.46 | 70.42±0.88 | 70.13±1.42 | 69.51±0.79 | 71.17±2.01 | 72.21±1.47 |
| MEPS21 | 71.38±0.20 | **72.02**±0.70 | 69.40±1.61 | 69.83±1.44 | 69.81±1.68 | 70.85±0.82 | 71.39±0.20 |
| STAR | **83.45**±3.09 | 83.17±3.47 | 82.89±1.51 | 81.22±2.55 | 81.22±3.60 | 83.03±2.07 | **83.45**±3.09 |
| BIO | 81.00±0.61 | 84.45±0.88 | 87.31±0.27 | 87.27±0.46 | **88.31**±0.72 | 84.20±0.70 | 81.00±0.61 |
| BLOG | 58.32±0.90 | 60.43±1.46 | **65.21**±0.58 | 59.03±1.03 | 54.55±0.77 | 55.76±1.26 | 58.32±0.90 |
| BIKE | 77.55±1.40 | 86.25±0.87 | **95.36**±1.32 | 94.23±1.40 | 95.06±1.06 | 84.65±1.85 | 77.55±1.40 |

## B.8 Additional Experiment Results

This section provides more experiment results. Additional visual results for the segmentation problem are also presented in Figure 5.

Table 15: Comparison of FFRAPS with the state-of-the-art method RAPS on Imagenet-Val. The FFRAPS method outperforms RAPS in most datasets.

| METHOD | ACCURACY | | COVERAGE | | LENGTH | |
|---|---|---|---|---|---|---|
| MODEL | TOP-1 | TOP-5 | RAPS | FFRAPS | RAPS | FFRAPS |
| RESNEXT101 | $0.793\pm0.001$ | $0.945\pm0.001$ | $0.908\pm0.002$ | $0.907\pm0.002$ | $2.012\pm0.035$ | $\mathbf{2.006}\pm0.039$ |
| RESNET152 | $0.784\pm0.001$ | $0.941\pm0.001$ | $0.909\pm0.003$ | $0.907\pm0.003$ | $2.144\pm0.034$ | $\mathbf{2.128}\pm0.058$ |
| RESNET101 | $0.774\pm0.001$ | $0.935\pm0.001$ | $0.906\pm0.004$ | $0.906\pm0.003$ | $2.348\pm0.151$ | $\mathbf{2.256}\pm0.037$ |
| RESNET50 | $0.761\pm0.001$ | $0.929\pm0.001$ | $0.907\pm0.004$ | $0.907\pm0.003$ | $\mathbf{2.560}\pm0.104$ | $2.594\pm0.069$ |
| RESNET18 | $0.698\pm0.001$ | $0.891\pm0.001$ | $0.906\pm0.003$ | $0.903\pm0.003$ | $4.560\pm0.147$ | $\mathbf{4.434}\pm0.168$ |
| DENSENET161 | $0.772\pm0.001$ | $0.936\pm0.001$ | $0.907\pm0.003$ | $0.907\pm0.002$ | $2.374\pm0.083$ | $\mathbf{2.328}\pm0.056$ |
| VGG16 | $0.716\pm0.001$ | $0.904\pm0.001$ | $0.904\pm0.002$ | $0.902\pm0.002$ | $3.566\pm0.098$ | $\mathbf{3.521}\pm0.065$ |
| INCEPTION | $0.696\pm0.001$ | $0.887\pm0.001$ | $0.903\pm0.003$ | $0.903\pm0.002$ | $5.410\pm0.350$ | $\mathbf{5.407}\pm0.133$ |
| SHUFFLENET | $0.694\pm0.001$ | $0.883\pm0.001$ | $0.902\pm0.001$ | $0.901\pm0.002$ | $5.001\pm0.121$ | $\mathbf{4.971}\pm0.073$ |

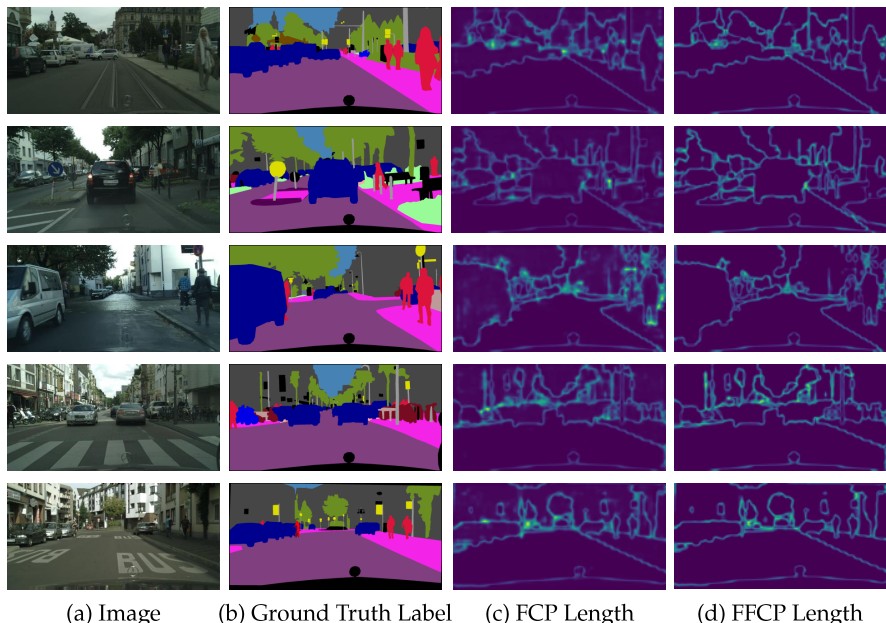

| (a) Image | (b) Ground Truth Label | (c) FCP Length | (d) FFCP Length |

Figure 5: Additional visualization results in segmentation task.

## B.9 Additional Experiments on Full CP

We further validate the generality of our proposed approach by applying it to the CV+ (Barber et al. [2021]) framework. Experiments were conducted on a synthetic multi-class classification dataset generated using scikit-learn. A total of 1,200 samples were created, with 1,000 used for training and 200 for testing. The dataset contained 10 features, including 3 informative and 2 redundant ones, distributed across 8 distinct classes, each consisting of a single cluster. Our method achieves higher efficiency while maintaining comparable coverage to the standard CV+ approach.

Table 16: Comparison of FFCV+ with CV+ on synthetic dataset.

| NUM | CLASS = 3 | | CLASS = 5 | | CLASS = 8 | |
|---|---|---|---|---|---|---|
| METHOD | COVERAGE | SIZE | COVERAGE | SIZE | COVERAGE | SIZE |
| CV+ | $90.50\pm0.01$ | $1.23\pm0.01$ | $90.00\pm0.01$ | $1.57\pm0.01$ | $92.00\pm0.01$ | $2.13\pm0.01$ |
| FFCV+ | $91.00\pm0.01$ | $1.20\pm0.01$ | $92.50\pm0.01$ | $1.55\pm0.01$ | $91.50\pm0.01$ | $2.11\pm0.01$ |

## B.10 Additional Experiments on Self-Supervised CP

SSCP (Seedat et al. [2023]) is a method that leverages self-supervised signals to enhance the adaptability and efficiency of conformal prediction intervals. As shown in Table 1, however, SSCP introduces substantially higher computational overhead due to the need to train two additional networks, making it approximately 2 to 50 times slower than FFCP. Across all settings, FFCP achieves shorter prediction bands, which may be attributed to SSCP's stronger dependence on the base model's predictive quality and the complexity of its auxiliary network training.

Table 17: Results on FFCP and SSCP

| METHOD | SSCP | | FFCP | |
|---|---|---|---|---|
| DATASET | TIME | LENGTH | TIME | LENGTH |
| SYNTHETIC | 7.18±0.99 | **0.25**±0.02 | 0.15±0.02 | 0.18±0.01 |
| COM | 1.33±0.16 | 2.52±0.14 | 0.03±0.01 | **1.84**±0.18 |
| MEPS19 | 10.86±0.76 | 5.32±0.33 | 0.14±0.01 | **3.13**±0.30 |
| STAR | 1.96±0.02 | 0.67±0.06 | 0.67±0.06 | **0.21**±0.01 |
| BIO | 13.49±3.81 | **1.53**±0.03 | 0.39±0.04 | 1.66±0.02 |
| BIKE | 5.92±0.03 | **0.73**±0.04 | 0.09±0.01 | 0.63±0.03 |

