# OpenReview forum: "Accelerating Feature Conformal Prediction via Taylor Approximation"
_NeurIPS.cc/2025/Conference — NeurIPS 2025 poster_

### Official Review · Reviewer_stTu · 2025-06-30

**Clarity:** 3
**Significance:** 1
**Originality:** 1
**Rating:** 4
**Confidence:** 4

**Summary:**

The authors propose an approximation scheme to map Conformal Prediction (PC) sets from the feature to the output space. The idea is to reweight the conformity score with the gradient of the feature map and define CP sets in the output space. The approach does not require extra training, produces intervals that are automatically marginally valid, and reduces efficiency gaps across different datasets.

**Questions:**

- What do you mean by *utilizing feature information in Vanilla CP*? How does this relate to approximating conditionally valid CP sets?
- What do you mean by *square condition* and *expanding* the differences between individual length quantiles in Theorem 4.2? How is *stability* in the feature and output spaces defined?
- Why are FCP and Vanilla's efficiencies so unstable in some cases?

**Ethical Concerns:**

["NO or VERY MINOR ethics concerns only"]

**Final Justification:**

After reading the authors' rebuttal, I raised my score to 4.

**Limitations:**

Limitations are not explicitly addressed. The paper does not have any potential negative societal impact.

**Quality:**

2

**Strengths And Weaknesses:**

**Strengths**
- The proposed conformity score does not require training and generates prediction sets that are automatically marginally valid.
- Compared to the vanilla and FCP approaches, the proposed conformity measure performs more consistently across datasets (see Table 2).

**Weaknesses**
- The paper is slightly incremental. While the context is new, approximating a nonlinear map with its linearization is a standard technique. The feature space map of [1] is also based on a linear relaxation.
- The intervals are computed in the output space. The proposed approach is vanilla CP with a task-specific conformity measure. In the original CP formulation, the conformity score is not required to be $|f(X) - Y|$ as assumed in Algorithm 1.
- The paper's motivations are the computational costs of computing output space prediction intervals and their non-exactness. It would be useful to expand the explanation on page 2 with some details on how the mapping technique of Xu et al. works and how the proposed approach improves it.

[1]
Teng, Jiaye, et al. "Predictive inference with feature conformal prediction." arXiv preprint arXiv:2210.00173 (2022).

---

> ### Author Rebuttal · Authors · 2025-07-30
>
> We would like to thank the reviewer  for the time and effort spent in reviewing our paper and we will address these concerns and provide our solutions in detail:
>
> **Weaknesses:**
>
> >The paper is slightly incremental. While the context is new, approximating a nonlinear map with its linearization is a standard technique. The feature space map of [1] is also based on a linear relaxation.
>
> We thank the reviewer for the insightful question. Here, we would like to clarify the motivation behind FFCP. We believe the contributions of FFCP lie primarily in two aspects:
>
> 1. FFCP, as a variant of FCP, **significantly reduces computational overhead while maintaining performance comparable to FCP**.
>
> 2. **FFCP exhibits strong extensibility. Its feature-level technique is largely orthogonal to most existing CP methods**, allowing it to be seamlessly integrated into classical frameworks such as CQR, RAPS, and LCP—as demonstrated in the experiments presented in this paper.
>
> It is true that linear relaxation is a widely used technique for approximating nonlinear operators. In the context of FCP [1], the incorporation of feature space information occurs primarily through two key steps:
> (a) mapping from the output space to the feature space; and
> (b) mapping back from the feature space to the output space.
>
> We believe the reviewer is referring to the linear relaxation used in step (b), which is implemented via LiPRA. Our experiments with FFCP suggest that leveraging feature space information can significantly benefit CP-based models. However, in practical deployment, we observed that
>
> Step (a) requires gradient descent to find the optimal feature representation.
>
> Step (b) relies on LiPRA for the backward mapping.
>
> Both procedures introduce considerable computational overhead. Our goal was to retain the benefits of using feature space information while substantially reducing computational cost, ideally without sacrificing predictive performance.
>
> To this end, we applied first-order Taylor approximations (which can indeed be viewed as a form of linear relaxation) to approximate both steps (a) and (b), thereby eliminating the need for gradient descent and LiPRA.
>
> We found that even such a simple approximation leads to strong empirical performance, comparable to that of the original FCP method. This finding reinforces the value of feature-space guidance in CP while highlighting a more efficient way to leverage it.
>
> >The intervals are computed in the output space. The proposed approach is vanilla CP with a task-specific conformity measure. In the original CP formulation, the conformity score is not required to be $|f(X)-Y|$ as assumed in Algorithm 1.
>
>
> As the reviewer correctly pointed out, FFCP constructs prediction sets in the output space. However, it is important to emphasize that, similar to FCP, **FFCP also leverages feature information.** From a constructional standpoint, FFCP follows the same conceptual pathway as FCP, mapping from the output space to the feature space and then back to the output space. By applying a first-order Taylor approximation during these two processes, FFCP achieves a significant simplification, resulting in an explicit and closed-form expression for the prediction band in the output space. This is a key advantage of FFCP over FCP: whereas FFCP allows for direct computation of the band length, FCP does not yield an explicit formulation, and thus its band length must be estimated through approximate procedures.
>
> We acknowledge that there are various possible formulations for computing non-conformity scores in Algorithm 1. We will revise the corresponding description in the next version to reflect this more accurately
>
>
> >The paper's motivations are the computational costs of computing output space prediction intervals and their non-exactness. It would be useful to expand the explanation on page 2 with some details on how the mapping technique of Xu et al. works and how the proposed approach improves it.
>
> We sincerely apologize for the confusion caused during reading. To clarify, we would first like to briefly explain the role of LiPRA (Xu et al.) in the context of our work.
>
> LiPRA is fundamentally a framework for computing neural network output bounds via linear relaxation-based perturbation analysis techniques, such as CROWN and DeepPoly. Within FCP, LiPRA is used to compute the image of a feature space ball in the output space. Specifically, given Algorithm 2 in FCP, we obtain a feature space set of the form  $\{g(v) : \|v - \hat{v}\| \leq Q_{1-\alpha}\}$, where $g$ is the prediction head, which is typically highly nonlinear. As a result, computing the corresponding band length in the output space lacks a closed-form expression.
>
> However, with LiPRA, given a trained neural network $g$, a reference point $\hat{v} = g(X)$, and a radius $Q_{1-\alpha}$ determined via FCP's Algorithm 2 , we can construct the set $\mathcal{B}_v$ = { $v: ||v-\hat{v}|| \leq Q$ }
> and LiPRA allows us to approximate the corresponding output set  $g(\mathcal{B}_v)$ = { $g(v) : v \in \mathcal{B}_v$ }.  This provides an approximation to the prediction band in the output space.
>
> In practice, however, this step introduces substantial computational overhead due to the complexity of the bound propagation and optimization procedures involved. To address this issue, FFCP adopts a first-order Taylor approximation to map the band length from the feature space to the output space. This significantly reduces computation while still preserving the core idea of leveraging feature-space uncertainty in conformal prediction.
>
>
>
> **Questions:**
> >What do you mean by utilizing feature information in Vanilla CP? How does this relate to approximating conditionally valid CP sets?
>
> In our work, utilizing feature information in Vanilla CP refers to our attempt to enhance Vanilla CP by incorporating feature-level information. Both FCP and FFCP can be viewed as extensions of Vanilla CP that leverage such information to improve performance.
>
> Regarding approximating conditionally valid CP sets, we would like to clarify that the FFCP technique is not specifically designed to address conditional validity. A typical approach for achieving conditional coverage is LCP (Localized Conformal Prediction). However, the feature-level technique introduced in FFCP can also be integrated into LCP to improve its performance. The resulting method, FFLCP, demonstrates improved conditional coverage properties. Detailed implementation and empirical results are provided in Section 5.3.2 and Appendix B.6 (Algorithm 4).
>
>
>
> >What do you mean by square condition and expanding the differences between individual length quantiles in Theorem 4.2? How is stability in the feature and output spaces defined?
>
> Due to our oversight, the statement of Theorem 4.2 was not clearly presented. The square conditions are intended to capture two key properties:
>
> 1. **Expansion**: We informally assume that the difference between the quantile and each individual score in the output space is smaller when measured in the feature space. This implies that performing the quantile operation (Step 4 in Algorithm 1 and Step 4 in Algorithm 2) does not compromise the effectiveness of the prediction bands, thereby supporting the efficiency of FFCP.
>
> 2. **Quantile Stability**: This condition ensures that the quantile computed on the calibration set generalizes well to the test set, which is essential for maintaining valid coverage.
>
> Under these two assumptions, we argue that the average band length of FFCP is theoretically smaller than that of Vanilla CP (split CP).
>
> To empirically validate the Expansion condition, we provide supporting experiments in Appendix B.2. In the case of FFCP, the difference between each sample and its corresponding quantile in the output space is the same as Vanilla CP. Therefore, Figure 4(a) can be interpreted as the distribution of non-conformity score deviations in the output space, while Figure 4(b) shows the same deviations measured in the feature space. The numerical results indicate that the deviations in the feature space are significantly smaller than those in the output space, thereby confirming that the Expansion condition is satisfied.
>
>
>
> >Why are FCP and Vanilla's efficiencies so unstable in some cases?
>
> We begin by analyzing the key factors that influence the performance of Vanilla CP, FCP, and FFCP:
>
> 1. Vanilla CP is the most basic method, and its performance primarily depends on a) the quality of the trained predictive model.
>
> 2. FCP is also affected by a)the quality of model training, but in addition, it is influenced by b)the optimization quality of the gradient descent procedure in FCP's Algorithm 2, and c)the quality of the LiPRA-based bound propagation.
>
> 3. FFCP (our method) does not rely on additional models or optimization procedures, so its performance depends solely on a)the quality of the predictive model.
>
> The band length produced by FCP is an approximation obtained via a two-step mapping from the output space to the feature space and back. This approximation introduces instability, which may result in degraded performance compared to Vanilla CP. For example, in Table 2, FCP performs worse than Vanilla CP on datasets like FB1 and BIKE, likely due to such instability.
>
> FFCP serves as an efficient approximation of FCP, achieving a substantial reduction in computational cost. Despite this simplification, FFCP achieves comparable performance to FCP. Notably, Table 2 shows that FFCP outperforms FCP in over half of the benchmark datasets.
>
> As FFCP relies on the output of a pre-trained model, it can also exhibit instability under certain conditions. In such rare cases, its performance tends to align with that of Vanilla CP.

---

> ### Author Response · Authors · 2025-08-05
>
> Dear Reviewer,
>
> Thank you again for your thoughtful feedback and for taking the time to review our work. We would like to check if there are any remaining questions or points that need further clarification. We are more than happy to provide additional information if needed.
>
> If you feel that our responses have adequately addressed the previous concerns, we would sincerely appreciate it if you could consider updating your overall evaluation of the paper.
>
> Thank you once again for your time and support.

---

> > ### Comment · Reviewer_stTu · 2025-08-06
> > **Thank you**
> >
> > Many thanks for all the clarifications and detailed answers. I raised my score to 4.

---

> > > ### Author Response · Authors · 2025-08-06
> > > **Thanks for feedback!**
> > >
> > > We sincerely thank you for your valuable comments and the improved rating. Your thoughtful suggestions have played an important role in helping us refine the paper. We will consider your clarifications and incorporate the discussion on limitations in the next revision. We truly appreciate your insights, which have significantly contributed to enhancing the overall quality of our work.

---

### Official Review · Reviewer_UCWd · 2025-06-30

**Clarity:** 2
**Significance:** 2
**Originality:** 2
**Rating:** 4
**Confidence:** 4

**Summary:**

This paper proposes to speed up the computation of Feature CP (FCP) via Taylor approximation. It then extends the proof of FCP to this new method, and carries out experiments comparing CP, FCP, and the proposal (Fast FCP).

**Questions:**

Presentation:
- the paper is hardly accessible to those who are not familiar with FCP.
- the denomination of "vanilla CP" is extremely confusing. The original CP is called transductive or ("full") CP, and if any CP method should be referred to as vanilla it should be that one. The authors compare against split (or "inductive") CP, and that's what the method should be called.
- Algorithm 1 is a special case of split CP, it is not exactly split CP (despite the description). In general, split CP is defined for an arbitrary nonconformity measure, which takes as input a bag of examples + a new example, and outputs a real number.
- According to the original definition of split CP, it would strongly seem that FCP is a special case of split CP, in that it uses a specific form of nonconformity measure. Can you clarify?
- The derivation between eq 5 and 6 should have been spelled out.


In classification problems, isn't eq (5) going to be "0" a lot of the times? Indeed, if $g(\hat{v}) = Y$, as one would hope most of the times for a decent classifier (features+head), then $\hat{v} \in \{v:g(v) = Y\}$ => eq 5 = 0. This means that "most of the times" the whole nonconformity measure computation can be skipped, leading to massive computational gains.

The empirical results are not as clearcut as the authors describe them. There are cases where FCP is better, cases where FFCP is better, and also cases were split CP is better.
In general, split CP is often one order of magnitude faster than FFCP, while its performance (length) isn't too far from it. The authors should consider rephrasing the final message to convey this.

Finally, the comparison should include Full CP, which should in principle achieve the best information coverage (split CP is on the other hand: it has the worst one). There are methods to speed up its computation, which should make it feasible to run this comparison [a.,b,c]

[a] Regression conformal prediction with nearest neighbours (Papadopoulos et al.)
[b] Exact Optimization of Conformal Predictors via Incremental and Decremental Learning (Cherubin et al.)
[c] Approximating Full Conformal Prediction at Scale via Influence Functions (Abad Martinez et al.)

**Ethical Concerns:**

["NO or VERY MINOR ethics concerns only"]

**Final Justification:**

Thank you for taking the time to write your detail rebuttal. I am satisfied with the promised changes, which I encourage the authors to include in their final version.

**Limitations:**

yes

**Quality:**

2

**Strengths And Weaknesses:**

Strengths:
- improves computational speed of FCP
- in some occasions, it is better than split CP.

Cons:
- presentation
- the results (method and theory) are relatively incremental

---

> ### Author Rebuttal · Authors · 2025-07-30
>
> Thanks to the reviewer for the thoughtful comments. We will provide a detailed explanation.
>
> **Questions:**
> >1.the paper is hardly accessible to those who are not familiar with FCP.
>
> We thank the reviewer for the correction and clarification of the concepts presented in our paper. Since FFCP is a variant of FCP, the current presentation may pose difficulties for readers unfamiliar with FCP. We will revise the relevant descriptions in future versions to improve clarity and minimize potential confusion.
>
> >2.the denomination of "vanilla CP" is extremely confusing. The original CP is called transductive or ("full") CP, and if any CP method should be referred to as vanilla it should be that one. The authors compare against split (or "inductive") CP, and that's what the method should be called.
>
> We sincerely apologize for the confusion caused in the reading. In future revisions, we will replace "Vanilla CP" with the more accurate term "split CP."
>
> >3.Algorithm 1 is a special case of split CP, it is not exactly split CP (despite the description). In general, split CP is defined for an arbitrary nonconformity measure, which takes as input a bag of examples + a new example, and outputs a real number.
>
> We agree that Algorithm 1 essentially corresponds to the canonical form of split conformal prediction.
>
> >According to the original definition of split CP, it would strongly seem that FCP is a special case of split CP, in that it uses a specific form of nonconformity measure. Can you clarify?
>
> Indeed, from the perspective of Full CP and split CP, FCP can be regarded as a special case of split conformal prediction, as it also partitions the dataset into a training set and a calibration set.
> The main innovation of FCP lies in how the non-conformity scores are computed during the calibration phase. In classical split CP, the non-conformity score is computed directly as $s = |Y - f(X)|$, and the prediction interval is constructed based on the quantile of these scores.
> In contrast, FCP similarly trains a model on the training set, but during calibration, it computes the non-conformity scores using Algorithm 2 presented in the paper. Specifically, it searches for an optimal feature representation that leads to a prediction close to the true label, and defines the score as $s_f = |v - h(X)|$. Then, the quantile $Q_{1-\alpha}$ of these scores is computed, which determines the conformal band radius.
> FCP further employs Band Estimation(incorporate techniques such as LiRPA) to estimate the output band length, and Band Detection (also using Algorithm 2) to evaluate coverage.
> Therefore, in essence, FCP is a variant of split CP, with the key distinction that it leverages information from the feature space in the computation of non-conformity scores.
>
>
>
> >The derivation between eq 5 and 6 should have been spelled out.
>
> Thanks for reviewer's question. We provide a detailed approximation procedure below:
>
> In FCP, the non-conformity score is originally defined as:
>
> $$
> \inf\_{v \in \{v : g(v) = Y\}} ||v - h(X)||.
> $$
>
> However, this is difficult to compute in practice. Therefore, FCP uses its Algorithm 2 to obtain a good approximation of the feature vector $v$, and then defines the score as:
>
> $$
> \text{score} = ||v - h(X)||.
> $$
>
> In Algorithm 2, the optimization objective is:
>
> $$
> \min_v \|g(v) - Y\|^2.
> $$
>
> We observe that directly optimizing over $v$ can be computationally complex. To simplify this, we consider a first-order Taylor expansion of $g(v)$ around $\hat{v} = h(X)$:
>
> $$
> g(v) \approx g(\hat{v}) + \nabla g(\hat{v})(v - \hat{v}).
> $$
>
> This implies:
>
> $$
> g(v) - g(\hat{v}) \approx \nabla g(\hat{v})(v - \hat{v}),
> $$
>
> and since we are reusing the FCP score form, we obtain:
>
> $$
> ||v - \hat{v}|| \approx \frac{|g(v) - g(\hat{v})|}{||\nabla g(\hat{v})||}.
> $$
>
> This gives an approximate closed-form for the non-conformity score, bypassing the need for gradient-based optimization during inference. Moreover, in a certain sense, FFCP can be viewed as equivalent to FCP when using Algorithm 2 with the number of gradient steps $M = 1$.
>
>
> >In classification problems, isn't eq (5) going to be "0" a lot of the times? Indeed, if $g(\hat{v})=Y$, as one would hope most of the times for a decent classifier (features+head), then $\{\hat{v}\in v:g(v)=Y\}$ => eq 5 = 0. This means that "most of the times" the whole nonconformity measure computation can be skipped, leading to massive computational gains.
>
> We thank the reviewer for the thoughtful question. Similar to FCP, FFCP is primarily designed for regression tasks, and the formulation presented in the main text reflects its application in the regression setting. **However, we would like to emphasize that the feature-level techniques of FFCP is highly extensible and largely orthogonal to most existing conformal prediction methods**, making it adaptable to a wide range of scenarios beyond regression.
>
> **For example, in classification tasks, we adopt an alternative version of our method, termed FFRAPS, which is better suited for classification.** The detailed implementation is provided in Section 5.3.1 and Appendix B.7, where Algorithm 5 illustrates how the feature-level techniques of FFCP are incorporated into classification tasks under the RAPS framework. In contrast, the version discussed in the main body of the paper is tailored for regression problems, where the goal is to estimate continuous-valued prediction bands.
>
>
> >The empirical results are not as clearcut as the authors describe them. There are cases where FCP is better, cases where FFCP is better, and also cases were split CP is better. In general, split CP is often one order of magnitude faster than FFCP, while its performance (length) isn't too far from it. The authors should consider rephrasing the final message to convey this.
>
> We understand the reviewer's concern, and we would like to offer the following clarifications:
>
> First, **the band length computed by FCP is inherently an approximation**, as it is derived by mapping a region from the feature space back to the output space using methods such as LiRPA. This process introduces potential instability in practice, which may lead to degraded performance compared to Vanilla CP. For instance, Table 2 shows that FCP underperforms on datasets like FB1 and BIKE due to such variability.
>
> Second, **FFCP can be viewed as an efficient approximation of FCP**. While it significantly reduces the computational burden, its performance remains comparable to that of FCP. In particular, as shown in Table 2, FFCP outperforms FCP on more than half of the evaluated datasets.
>
> Lastly, **since FFCP relies on the quality of the trained model**, it may exhibit instability in rare cases. In such scenarios, its performance is generally on par with that of Vanilla CP.
>
>
>
> >Finally, the comparison should include Full CP, which should in principle achieve the best information coverage (split CP is on the other hand: it has the worst one). There are methods to speed up its computation, which should make it feasible to run this comparison [a.,b,c]
>
> We sincerely thank the reviewer for the thoughtful suggestion. Since FFCP is designed as a fast variant of FCP, and FCP itself can be viewed as a specific instance of split CP, our comparisons have primarily focused within the split CP framework.
>
> The main reason we did not include comparisons with full CP methods is due to their substantial computational overhead. Full CP typically requires retraining or refitting the model multiple times, which becomes impractical for complex tasks such as image segmentation or image classification. Therefore, implementing full CP in such settings poses significant challenges.
>
> That said, we fully agree that including full CP in standard regression settings would provide a more comprehensive and objective comparison across CP methods. We will include these results in the next version of our paper.

---

> ### Author Response · Authors · 2025-08-05
>
> Dear reviewer,
>
> We would like to check if there are any further concerns about our article and rebuttal. We are glad to answer additional questions if there is anything unclear. Also, we humbly hope you could consider updating the overall rating if the previous issues have been fixed.
>
> Thank you for your support.

---

> > ### Comment · Reviewer_UCWd · 2025-08-06
> >
> > Thank you for taking the time to write your detail rebuttal. I am satisfied with the promised changes, which I encourage the authors to include in their final version.

---

> > > ### Author Response · Authors · 2025-08-06
> > > **Thanks for feedback!**
> > >
> > > Thank you very much for your generous feedback and the updated score. We are especially grateful for the constructive points you raised, which have played a key role in shaping our revisions. We will carefully address your clarifications and ensure that the discussion on limitations is expanded in the revised version. Your feedback has been extremely valuable to our work, and we sincerely appreciate your contribution to improving the clarity and rigor of the paper.

---

### Official Review · Reviewer_JJoX · 2025-07-03

**Clarity:** 4
**Significance:** 3
**Originality:** 1
**Rating:** 4
**Confidence:** 3

**Summary:**

The paper proposes a variation of Feature Conformal Prediction (FCP), which they call Fast Feature Conformal Prediction (FFCP). The core idea is to approximate the prediction head with its linearization, obtained as a first order Taylor expansion. The key benefit of this approximation is efficiency and speed improvement, with the linear model the non-conformity score can be computed analytically for a fraction of the cost of standard FCP. The drawback of this approximation is a potential decrease in performance, but the authors claim that such performance decrease is neglectable and support the claim through extensive empirical evaluation.

**Questions:**

-Can the author discuss how is it possible that the faster linarized model with FFCP in some settings performs better that the original not-linearized model with FCP? I find this effect very counterintuitive. Do the authors have some explanation of this effect?

-Why didn't the authors include the code for reproducibility in the supplementary material?

**Ethical Concerns:**

["NO or VERY MINOR ethics concerns only"]

**Limitations:**

yes

**Paper Formatting Concerns:**

I didn't noticed any major formatting issues

**Quality:**

3

**Strengths And Weaknesses:**

The paper is well written an easy to read. The concept are well introduced and the proposed method is well explained. However, I think that the first order Taylor approximation of the prediction head should be made more explicit, so far it is only clarified in line 162-163, but this is a very fundamental step of your proposed method. For example you can modify line 8 of the abstract from "leveraging a Taylor expansion" to "leveraging a first-order Taylor expansion". Despite this and some minor comments I list below, I overall like the paper.

The idea is not very complex or novel (it somehow resemble the step from Laplace Approximation to Linearized Laplace approximation). Nonetheless the advantages are clear and well supported empirically: a faster method that does not decrease performance. For this reason I'm inclined for acceptance.

My only main concern is about the performances of FFCP compared to FCP. It is cool that the performance do not degrade much, which is a good motivation for the faster method. However, I find it quite surprising that, in some settings, the performance do actually increase for the faster linearized method. I would expect the performance to at most stay the same when the first order Taylor approximation is applied.

Some minor comments that I'd like to be addressed:

-In line 505 you write "where H is any function". This sound too strong and honestly feels wrong. Are you sure that you didn't miss some regularity conditions?

-in Line 130, does it mean that the set of indexes $I=[n]$? Or am I missing something here?

-in Line 204 and 205 there is a clear repetition that could be avoided.

-in Line 495 there is a typo: "which adjusted" should be "which is adjusted"

-in Line 505 there is a typo: "then the features" should be "then if the features", right?

---

> ### Author Rebuttal · Authors · 2025-07-29
>
> We are grateful for the reviewer's valuable feedback and recognition of our work. We will provide additional explanations below.
>
>
> **Weaknesses:**
> >For example you can modify line 8 of the abstract from "leveraging a Taylor expansion" to "leveraging a first-order Taylor expansion".
>
> We apologize for the confusion caused and truly appreciate the reviewer’s clarification. As you correctly pointed out, our method is based on a first-order Taylor expansion, and we agree that the description should be more precise and rigorous. We will revise the corresponding statements in the next version to avoid any misunderstanding.
>
> >My only main concern is about the performances of FFCP compared to FCP. It is cool that the performance do not degrade much, which is a good motivation for the faster method. However, I find it quite surprising that, in some settings, the performance do actually increase for the faster linearized method. I would expect the performance to at most stay the same when the first order Taylor approximation is applied.
>
> >Can the author discuss how is it possible that the faster linarized model with FFCP in some settings performs better that the original not-linearized model with FCP? I find this effect very counterintuitive. Do the authors have some explanation of this effect?
>
> We believe this phenomenon arises from the fact that the band length producted by FCP is not an exact value. Since FCP computes the band in the feature space and then maps it back to the output space, the resulting length is inherently an approximation rather than a directly computed quantity. Specifically, FCP relies on gradient descent during the calibration phase to identify optimal feature representations and employs LiRPA techniques during the test phase. Both procedures involve approximations and can introduce substantial variability. As shown in our experimental results, FCP may even underperform Vanilla CP in certain cases. The performance of FCP is highly sensitive to the number of gradient descent iterations and the choice of hyperparameters in LiRPA, both of which require careful tuning.
>
> In contrast, FFCP circumvents the costly bidirectional mapping steps and features an explicit computational structure in the output space, resulting in greater stability. This often allows FFCP to outperform FCP in terms of predictive performance under practical settings. However, we acknowledge that if optimal parameters for FCP’s optimization steps can be identified, it typically achieves the best coverage-performance trade-off across most scenarios. In this sense, FCP remains superior on average in terms of predictive guarantees, **while FFCP's primary advantage lies in its substantial reduction in computational cost.**
>
>
>
> >In line 505 you write "where H is any function". This sound too strong and honestly feels wrong. Are you sure that you didn't miss some regularity conditions?
>
> We apologize for the confusion caused by our oversight. To clarify, $H$ refers to $\mathcal{H}(v, X)$, which denotes the length of the prediction interval in the output space for a sample $X$, given a length $v$ in the feature space. Specifically, for FFCP, this is defined as $\mathcal{H}(v, X) = Q_{1-\alpha} \cdot \| \frac{\partial f(X)}{\partial h(X)} \|$. Thus, $\mathcal{H}$ is not an arbitrary function; rather, it has a well-defined and principled form. We will revise the description in the next version to improve clarity and correctness.
>
> >in Line 130, does it mean that the set of indexes $[n]$? Or am I missing something here?
>
> We apologize for the confusion，this should indeed be $\mathcal{I}$, and the use of $n$ is a typo.
>
> >in Line 204 and 205 there is a clear repetition that could be avoided
> >in Line 495 there is a typo: "which adjusted" should be "which is adjusted"
> >in Line 505 there is a typo: "then the features" should be "then if the features", right?
>
> We sincerely thank the reviewer for their careful and thorough review, and for dedicating substantial time and effort to our paper. The issues mentioned above were caused by our oversight, and we will promptly correct these errors.
>
> **Questions:**
> >Why didn't the authors include the code for reproducibility in the supplementary material?
>
> We regret any confusion this may have caused. We have organized the codebase on GitHub and will make it publicly available in the next version.

---

> > ### Comment · Reviewer_JJoX · 2025-08-08
> >
> > I thank the authors for addressing my comments.
> > I'm still partially concerned about the lack of code (and consequently about the transparency of the experiments), so I keep my score.

---

> > > ### Author Response · Authors · 2025-08-08
> > > **Thanks for feedback!**
> > >
> > > We sincerely thank the reviewer for their time and effort in reviewing our manuscript. We also commit to making our code publicly available. Thank you again for your support.

---

> ### Comment · Area_Chair_3C5m · 2025-08-07
>
> Dear Reviewer JJoX,
>
> Can you reply to authors' rebuttal?
>
> AC

---

### Official Review · Reviewer_4qMa · 2025-07-06

**Clarity:** 3
**Significance:** 2
**Originality:** 2
**Rating:** 4
**Confidence:** 4

**Summary:**

the paper introduces Fast Feature Conformal Prediction (FFCP), an taylor approximation version of prior work Feature Conformal Prediction (FCP).  By using a non-conformity score that uses the gradient of the decoder (part of NN after the feature space embedding used for UQ),  FFCP simplifies the non-linear operations involved in transforming confidence bands from feature space to output space, which is a bottleneck in FCP.  The authors provided theoretical justifications and introduce extensions of FFCP to other CP algorithms.

**Questions:**

Experiment section

- why is the performance not compared to other feature-based scaled CP methods? You cite Seedat et al 2023 and 2024 - the reason of it needing extra training does not excuse the lack of comparison. One can argue that once amortized, the time consumption of Seedat 2023's method is faster than FCP. If completely without comparison to other methods, where does the reader place this method in the literature?

- "For FFCP, we select the shortest band length among all layers." this is unfair comparison. the authors should choose one layer and stick with it.

- why does Vanilla CP has the exact same numbers as FFCP on the synthetic, star, and bio datasets in table 2? They have different numbers even in table 6.

Unsupported Claims

- even just comparing to FCP, it is unclear from table 2 that FFCP produces "comparable" quality uncertainty sets. FFCP's intervals some times 50%-100% wider.

- I'm not sure Figure 2 shows the qualitative result supporting your claim of "the FFCP producing more refined and sharper uncertainty bands.", as the presentation makes the difference very subtle. Moreover, what do you mean by : "FCP operates at the image level, ensuring coverage for the entire predicted mask as a whole, while FFCP adopts a coordinate-wise approach, offering per-pixel statistical guarantees that better reflect local uncertainty."? Aren't the proven theoretical guarantees the same?


----
I'm happy to raise my score if my concerns are addressed.

**Ethical Concerns:**

["NO or VERY MINOR ethics concerns only"]

**Final Justification:**

the main concern with limited contribution and  limitations discussion remains, but my concerns with the legitimacy of experiments have been addressed.

**Limitations:**

The limitations of the method sparingly discussed, and are not informatively summarized in the discussion/conclusion. Some questions I would appreciate discussions are:

- What are the failure cases of the method?
- How does it compare to other locally-scaled CP methods in different data setups?
-  to what extent are the expressivity of FCP limited in FFCP?  (to my understanding FFCP cannot create disjoint prediction intervals for regression tasks but FCP can?)

**Quality:**

2

**Strengths And Weaknesses:**

Strengths

- the largest contribution and motivation of the paper is practicality. the authors improve the FCP algorithm such that it bypasses its biggest time bottleneck, LiPRA projection, while retaining most of its strengths.

-  the method is valid and the theory is sound (although heavily building on Teng et al. [2022]).

- the extensions to CQR and RAPS is nice, bringing more usability to the method. (I think there are a lot of potential to section 5.3, but it needs more analysis and organization to integrate to the main paper.)

- overall the paper is clearly written and easy to understand.

----
Weaknesses

- insufficient contribution. the paper essentially proposes a faster (first degree taylor) approximation to an existing method, which is relatively straightforward. The effectiveness of the algorithm (compared to split CP) highly relies on how accurate the taylor approximation is, where there are no guarantees. Neither the theory or findings brings new insights to CP in my opinion; I'm not sure the contribution is sufficient for a top conference like neurips.

- the experiment results are less than comprehensive and need more analysis. please see questions section for details.

- some unsupported claims in the paper. please also see questions section.

- i think calling the original split conformal prediction algorithm as "vanilla" is a bit derogatory and unprofessional/not rigorous. I would just use "split CP" in its place; people will know what you are talking about.

---

> ### Author Rebuttal · Authors · 2025-07-30
>
> We appreciate the reviewer's insightful comments. We will address these issues one by one.
>
> **Weaknesses:**
> >insufficient contribution. the paper essentially proposes a faster (first degree taylor) approximation to an...
>
> We understand the reviewer's concern, but we would like to clarify that the core advantages of FFCP lie in two aspects:
>
> 1. As a variant of FCP, FFCP significantly reduces computational cost while effectively leveraging feature space information. As a result, it consistently outperforms Vanilla CP (i.e., split CP) and achieves performance comparable to FCP.
>
> 2. The feature-level technique in FFCP is highly extensible and largely orthogonal to existing CP methods. It can be readily integrated into various frameworks, including CQR, RAPS, and LCP—and consistently improves performance, as demonstrated in our experiments.
>
> While our method leverages a first-order Taylor expansion, it aims to address the core challenge of efficiently utilizing feature space information. FCP improves predictive performance by incorporating feature representations but suffers from high computational costs due to gradient descent and LiPRA. In contrast, FFCP approximates the feature mapping using gradient information, avoiding costly computations while retaining the benefits of feature-guided conformal prediction. It also provides a closed-form prediction band, improving interpretability.
>
> From the perspective of extensibility, we have applied the FFCP technique to CQR, RAPS, and LCP (among others), and observed consistent performance improvements over the original models.
>
> >i think calling the original split conformal prediction algorithm as "vanilla" is a bit ...
>
> Thanks for the reviewer’s reminder, we will make the necessary corrections in the next revision.
>
> **Questions:**
> Experiment section
> >why is the performance not compared to other feature-based scaled CP methods? You cite Seedat et al...
>
> Thanks for your constructive suggestion, we have supplemented the results by including those from the referenced literature.
> #### Table A：Results of SSCP and FFCP
>
>
> | Dataset   | Time (s)         |           | SSCP Length       | FFCP Length       |
> |-----------|------------------|-----------|-------------------|-------------------|
> |           | SSCP             | FFCP      |                   |                   |
> | SYNTHETIC | 7.18±0.99        | 0.15±0.02 | 0.25±0.02         | 0.18±0.01         |
> | COM       | 1.33±0.16        | 0.03±0.01 | 2.52±0.14         | 1.84±0.18         |
> | MEPS19    |10.86±0.76        | 0.14±0.01 | 5.32±0.33         | 3.13±0.30         |
> | STAR      | 1.96±0.02        | 0.67±0.06 | 0.67±0.06         | 0.21±0.01         |
> | BIO       |13.49±3.81        | 0.39±0.04 | 1.53±0.03         | 1.66±0.02         |
> | BIKE      | 5.92±0.03        | 0.09±0.01 | 0.73±0.04         | 0.63±0.03         |
>
> In terms of runtime, SSCP incurs substantially higher computational overhead, requiring the training of two additional networks, and is approximately 2 to 50 times slower than FFCP. In all cases, FFCP also achieves shorter band length. This may be attributed to SSCP’s stronger reliance on the base model’s predictive quality and the effectiveness of its auxiliary network training.
>
>
>
> >For FFCP, we select the shortest band length among all layers."...
>
> Thanks for the reviewer’s question. This may be due to our lack of clarity in the paper. FFCP method records the total computation time for each layer, and then selects the layer that achieves the best results.
>
> >why does Vanilla CP has the exact same numbers as FFCP on ...
>
> We sincerely apologize for the confusion caused by our unclear presentation, and we would like to take this opportunity to clarify the issue.First, regarding FFCP, the prediction band takes the form in Eq (8), where $Q_{1-\alpha}=\{\|f(X)-Y\|/\|\nabla g(v)\|\}$ When we evaluate the final (top) layer, the prediction function satisfies $g(\cdot)=f(\cdot)$, and hence $\nabla g(v)=\frac{\partial f(X)}{\partial g(v)}=\frac{\partial f(X)}{\partial f(X)}=\mathbf{1}$. As a result, in this case, FFCP degenerates to the same form as Vanilla CP, meaning their performance is equivalent. This explains why the results of FFCP and Vanilla CP are identical on certain datasets in Table 2.
>
> Table 6 presents a different setting where the base model is under-trained. In this case, feature representations are less reliable, and FFCP tends to perform similarly—or even worse—than Vanilla CP. Due to our oversight, only results from Layer 3 were reported. We have updated the results in Table B by selecting the best-performing layer, which confirms that FFCP underperforms when the model is not sufficiently trained.
>
> #### Table B: Untrained model Length on FFCP \(full version\)
> |Dataset|Layer1|Layer2|Layer3|Layer4|Layer5 \(Vanilla\)|
> |---|---|---|---|---|---|
> |SYNTHETIC|2.87±0.03|2.81±0.03|2.89±0.01|2.41±0.01|**2.34±0.01**|
> |COM|5.41±0.14|5.30±0.18|5.03±0.09|**4.73±0.08**|4.86±0.13|
> |FB1|3.90±0.01|3.77±0.10|3.62±0.08|3.5699±0.08|**3.5697±0.09**|
> |FB2|4.02±0.09|3.90±0.8|3.74±0.08|3.6625±0.06|**3.6624±0.11**|
> |MEPS19|4.38±0.06|4.41±0.07|4.42±0.07|4.38±0.07|**4.33±0.07**|
> |MEPS20|4.43±0.21|4.44±0.23|4.48±0.25|4.46±0.25|**4.41±0.23**|
> |MEPS21|**4.4081±0.18**|4.46±0.18|4.48±0.18|4.4143±0.15|4.4133±0.17|
> |STAR|2.49±0.06|2.40±0.05|2.15±0.03|1.94±0.01|**1.88±0.01**|
> |BIO|4.27±0.02|4.14±0.02|4.07±0.02|**4.04±0.02**|4.08±0.02|
> |BLOG|2.56±0.15|2.54±0.15|2.57±0.13|2.55±0.14|**2.53±0.12**|
> |BIKE|4.69±0.09|4.67±0.07|4.57±0.09|4.60±0.10|**4.56±0.09**|
>
> Unsupported Claims
> >even just comparing to FCP, it is unclear from table 2 ...
>
> As the reviewer correctly pointed out, FFCP does exhibit slightly inferior performance compared to FCP in a few cases. However, its primary advantage lies in the substantial reduction of computational cost, making it a practical trade-off between runtime efficiency and predictive performance.
> Within our framework, we define "comparability" as the ability of FFCP to either outperform or closely match the performance of FCP across a wide range of datasets. As shown in Table 2, FFCP achieves superior performance on more than half of the evaluated benchmarks.
>
> Furthermore, unlike FCP, which estimates band length via an implicit and approximate procedure that may lead to instability, FFCP provides a closed-form expression for the prediction band. This enables exact and efficient computation, enhancing both interpretability and robustness.
> For these reasons, we argue that directly comparing the band lengths of FCP and FFCP on a single dataset may be misleading. A more appropriate evaluation should consider the relative performance across multiple datasets to assess the overall effectiveness and efficiency of FFCP.
>
>
>
> >I'm not sure Figure 2 shows the qualitative result...
>
> Apologies for the confusion. In Figure 2, our intention was to illustrate that FFCP produces visually sharper edges in segmentation tasks, rather than claiming “sharper uncertainty bands.” We will revise this statement in the next version of the paper to avoid ambiguity.
>
> Regarding the quantile computation, FCP adopts an image-level approach, where each image is treated as a single data point during calibration，and the $\infty$-norm $||Y-\hat{Y}||\_\infty=\max\_{i \leq d}\|y\_i-\hat{y}\_i\|$ is used to define the non-conformity score. In contrast, FFCP computes quantiles in a pixel-level manner, as highlighted in Remark 1, treating each pixel as an individual data instance.
>
> We believe that uncertainty estimation in image segmentation is more appropriately measured at the pixel level. Therefore, we updated the quantile computation accordingly. This change led to clearer visual boundaries in the prediction masks. (For completeness, we also experimented with applying FFCP under the image-level quantile computation, but the results were suboptimal.)
>
> **Limitations:**
> >What are the failure cases of the method?
>
> We sincerely thank the reviewer for their comprehensive evaluation of our paper. In the case of FFCP specifically, since the calibration step involves division by the gradient norm, issues may arise when the gradient is very small. We recognize these limitations and view them as important directions for future improvement of our work.
>
> >How does it compare to other locally-scaled CP methods in different data setups?
>
> We expand on LCP-related models in Appendix B.6, showing that the feature-level technique of FFCP can be extended to other CP frameworks. For instance, we introduce FFLCP as an extension of LCP for group coverage, and Table 12 demonstrates that FFLCP consistently outperforms LCP across datasets.
> Overall, the FFCP feature-based approach is largely orthogonal to existing CP methods, making it a broadly applicable enhancement for improving performance via feature information.
>
> >to what extent are the expressivity of FCP limited in FFCP? ...
>
> We believe that both FCP and FFCP produce *connected* prediction regions in regression tasks. This stems from two key observations:
> (1) the feature space region is typically assumed to be bounded.
> (2) the prediction head $g$ is a continuous function.
>
> Under these assumptions, the pre-image set $\{ v : g(v) \in \mathcal{C} \}$ for a connected set $\mathcal{C} \subset \mathbb{R}^d$ (such as a prediction interval) remains connected.
> While FCP may allow more flexible or complex *shapes* within the connected region due to the implicit nature of its band estimation (e.g., via LiPRA), FFCP yields prediction intervals with explicit structure resulting from its first-order Taylor approximation. This structural constraint limits the expressivity of FFCP compared to FCP, but it also brings substantial computational benefits and interpretability. Thus, neither FCP nor FFCP produces disjoint prediction intervals in standard regression settings.

---

> > ### Comment · Reviewer_4qMa · 2025-08-05
> > **thank you for the rebuttal**
> >
> > Thank you for the clarification, my questions have been addressed and I have increased my scores accordingly.
> >
> > For the finalized paper, i hope these clarification can be included, with an added discussion on limitations.

---

> > > ### Author Response · Authors · 2025-08-05
> > > **Thanks for feedback!**
> > >
> > > We deeply appreciate your thoughtful feedback and revised rating!
> > > We truly appreciate your constructive feedback, which has undoubtedly contributed to improving the quality of our paper. We will consider your clarifications and include the added discussion on limitations in the next version. Once again, thank you for your insightful input, which has helped enhance the overall quality of our work!

---

### Comment · Area_Chair_3C5m · 2025-08-04

Dear Reviewers,

Please review the authors' rebuttal and update your evaluation as needed.

AC

---

### Note · Authors · 2025-08-14

Dear Area Chairs and Reviewers,

We sincerely thank you for the time and effort you devoted to reviewing our submission. Your constructive feedback has played a vital role in improving the clarity and completeness of our work. We appreciate the positive recognition our paper received after the rebuttal and would like to take this opportunity to concisely reiterate the strengths and contributions of our proposed method, Fast Feature Conformal Prediction (FFCP).

During the review process, three primary concerns were raised:

Q1: What is the core advantage of FFCP? (Reviewers 4qMa, stTu)

A1: FFCP is a computationally efficient variant of FCP that retains the ability to incorporate feature information while significantly accelerating the conformal prediction process. Specifically, FFCP achieves over 50× speedup by leveraging a first-order Taylor expansion to simplify the bidirectional mapping between feature and output spaces inherent in FCP. Despite this simplification, FFCP maintains comparable empirical performance.

Additionally, FFCP exhibits strong modularity and extensibility. The feature-utilization technique proposed in FFCP is orthogonal to most conformal prediction frameworks, allowing easy integration into various predictive models. In this work, we demonstrate its applicability to CQR, LCP, and RAPS, highlighting its versatility across multiple settings.

Q2: How does FFCP compare with FCP and vanilla CP (Split CP)? (Reviewers 4qMa, JJoX, UCWd)

A2: We showed across various datasets that FFCP matches or outperforms FCP in most cases. In rare instances where FFCP underperforms relative to vanilla CP, we attribute this to variability in base model performance rather than limitations of FFCP itself.

Q3: How does FFCP perform relative to other state-of-the-art methods? (Reviewers 4qMa, UCWd)

A3: In the rebuttal, we provided comparative results with SSCP, where FFCP consistently achieved better or comparable results. We also commit to including comparisons with Full-CP in the final version.

We are immensely grateful for the reviewers’ engagement with our work and their insightful comments. We hope this summary provides a clear and concise perspective on the contributions of FFCP and look forward to your final evaluation.

Sincerely,

The Authors

---

### Decision · Program_Chairs · 2025-09-17

**Decision:**

Accept (poster)

**Comment:**

This paper presents a well-motivated and sound method to significantly accelerate feature conformal prediction using a Taylor approximation, a contribution reviewers found to be practical and valuable. The paper is clearly written, and the proposed approach achieves a notable computational speedup while maintaining comparable performance to the original, more computationally intensive method.
While reviewers initially raised valid concerns about the experimental comparisons and reproducibility, the authors provided a thorough rebuttal with additional results and clarifications that addressed most of these issues. The constructive dialogue significantly strengthened the submission, with the authors clarifying key technical points and committing to release their code, bolstering confidence in the work's contributions to the field.